



# Comparison of formaldehyde tropospheric columns in Australia and New Zealand using MAX-DOAS, FTIR and TROPOMI

Robert G. Ryan[1,2], Jeremy D. Silver[1], Richard Querel[3], Dan Smale[3], Steve Rhodes[4], Matt Tully[4], Nicholas Jones[5], and Robyn Schofield[1,2,6]

[1]School of Earth Sciences, The University of Melbourne, Melbourne, Australia
[2]ARC Centre of Excellence for Climate System Science, Sydney, Australia
[3]National Institute for Water & Atmosphere Research, Lauder, New Zealand
[4]Bureau of Meteorology, Melbourne, Australia
[5]School of Earth, Atmospheric & Life Sciences, University of Wollongong, Australia
[6]ARC Centre of Excellence for Climate Extremes, Sydney, Australia

**Correspondence:** Robert Ryan
(rgryan92@gmail.com)

**Abstract.** South-eastern Australia has been identified by modelling studies as a hotspot of biogenic volatile organic compound (VOC) emissions, however long term observational VOC studies are lacking in this region. Here, two and a half years of MAX-DOAS formaldehyde (HCHO) measurements in Australasia are presented, from Broadmeadows in northern Melbourne, Australia and from Lauder, a rural site in the South Island of New Zealand. Across the measurement period from December 2016 to

November 2019, the mean formaldehyde column measured by the MAX-DOAS at Lauder was $2.50 \pm 0.61 \times 10^{14}$ molec.cm$^{-2}$ and at Broadmeadows was $5.40 \pm 1.59 \times 10^{15}$ molec.cm$^{-2}$. In both locations the seasonal cycle showed a pronounced peak in Austral summer (DJF) consistent with temperature-dependent formaldehyde production from biogenic precursor gases. The amplitude of the seasonal cycle at Lauder was $0.7 \times 10^{15}$ molec.cm$^{-2}$ while it was $2.0 \times 10^{15}$ molec.cm$^{-2}$ at Broadmeadows. The Lauder MAX-DOAS HCHO measurements are compared with 27 months of co-located fourier-transform infrared (FTIR)

observations. The seasonal variation of Lauder MAX-DOAS HCHO, smoothed by the FTIR averaging kernels, correlated strongly with the FTIR measurements, with linear regression slope of 0.91 and R$^2$ of 0.81 for monthly averaged formaldehyde partial columns. In addition to ground-based observations, a clear way to address the VOC measurement gap in areas such as Australasia is with satellite measurements. Here we demonstrate that the Tropospheric Monitoring Instrument (TROPOMI) can be used to distinguish formaldehyde hotspots in forested and agricultural regions of south-eastern Australia. The MAX-

DOAS measurements are also compared to TROPOMI HCHO vertical columns at Lauder and Melbourne; very strong monthly average agreement is found for Melbourne (regression slope of 0.61, R$^2$ of 0.95) and a strong agreement is found at Lauder (regression slope of 0.73, R$^2$ of 0.61) for MAX-DOAS vs TROPOMI between May 2018 and November 2019. This study, the first long term satellite comparison study using MAX-DOAS in the southern hemisphere, highlights the improvement offered by TROPOMI's high resolution over previous satellite products and provides the groundwork for future studies using ground

based and satellite DOAS for studying VOCs in Australasia.



# 1 Introduction

Volatile organic compounds (VOCs) influence tropospheric oxidation chemistry in three key ways: oxidation by OH (thereby decreasing oxidation capacity), formation of OH (e.g. by photolysis of formaldehyde) and contributing to tropospheric ozone formation. Remote sensing of oxidised VOCs such as formaldehyde has been demonstrated as an effective method for constraining VOC emissions and for studying the role of VOCs in atmospheric reactivity (see Kefauver et al. (2014) and references therein).

Formaldehyde has atmospheric mixing ratios ranging from several hundred parts per trillion (ppt) in unpolluted marine air (Mahajan et al., 2010; Peters et al., 2012) to tens of parts per billion (ppb) in polluted urban air (e.g. Zhu et al., 2017). Primary sources of formaldehyde include direct emission from fossil fuel combustion and wild fires. The main secondary sources of HCHO are oxidation of methane, isoprene and monoterpenes. Methane is considered to be the primary background HCHO source globally (Pfister et al., 2008), and because it is a potent greenhouse gas, studying background formaldehyde levels has important climate change implications. Isoprene and monoterpenes emitted from vegetation constitute the main source of biogenic carbon to the atmosphere (Guenther et al., 2012). While methane is considered the most important OH sink in background oceanic air, over land isoprene and monoterpenes constitute the largest OH reactivity and hence these biogenic VOCs play a crucial role in determining oxidative capacity (Fuentes et al., 2000; Lelieveld et al., 2008). Isoprene and monoterpenes are also thought to play a strong role in the climate system through radiative forcing by secondary formation of organic aerosols (Henze et al., 2008). Photolysis and reaction with OH and photolysis limit the lifetime of formaldehyde to several hours during the daytime which facilitates the comparison of co-located measurements and also means that spatially resolved HCHO measurements closely resemble the distribution of its VOC sources (Zhu et al., 2016).

Biogenic VOC emissions in Australasia are among the highest in the world due to the abundance of Australian endemic eucalyptus trees, known to be high isoprene and monoterpene emitters (Winters et al., 2009; Guenther et al., 2012). Global scale modelling has suggested Australia has the highest isoprene-derived formaldehyde levels of any other continent (Pfister et al., 2008), however constraining biogenic VOC emissions has so far proven challenging in Australia. Formaldehyde measurements, for example from satellites, are common proxies for biogenic VOC emissions but the accuracy of this under low $NO_x$ conditions has not been observationally verified (Zhu et al., 2016; Wolfe et al., 2016), likely due to uncertainties in differentiating HCHO from different anthropogenic, isoprene and monoterpene sources. Emmerson et al. (2016, 2018) highlighted this by demonstrating that the MEGAN biogenic emissions scheme, used in numerous global and regional-scale chemistry/climate models, overestimates isoprene and underestimates monoterpenes in the thickly eucalyptus-forested south-east of Australia. Therefore, reliable, long term biogenic VOC measurements are needed in the Australasian region.

The Multi-axis differential optical absorption spectroscopy (MAX-DOAS) technique, a passive spectroscopic method which use scattered solar radiation, can facilitate this through measurement of formaldehyde. In the last decade HCHO MAX-DOAS measurements have been reported from many locations in the world (Hoque et al., 2018a, b; Heckel et al., 2005; Pinardi et al., 2013; Peters et al., 2012; Vigouroux et al., 2009), however none so far have been reported in Australasia.





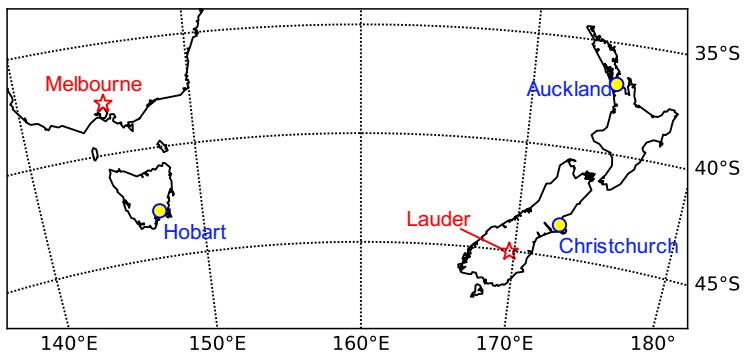

**Figure 1.** Map of New Zealand and south-eastern Australia showing the locations of Lauder and Melbourne (red stars) in relation to each other and to Hobart, Christchurch and Auckland (blue and yellow points).

Developments in satellite sensors and retrievals of atmospheric trace gases over the past two decades can offer new insights
into air quality and composition (Martin, 2008). Validation by ground-based instrumentation is an important step in understanding the utility of such satellite data products. Because satellite instruments and MAX-DOAS share the same spectroscopic technique for retrieving UV and visible absorbing trace gases, MAX-DOAS is an ideal validation tool as demonstrated for HCHO in several previous papers (e.g. Chance et al., 2000; Thomas et al., 1998; Hoque et al., 2018b; De Smedt et al., 2015; Vigouroux et al., 2009; Lee et al., 2015; Kurosu et al., 2007). However no such validation studies have so far been published
for the Australasian region.

Measurements in two locations are discussed in this paper: Broadmeadows, on the northern fringe of Melbourne in south-eastern Australia and Lauder, a remote locality in the South Island of New Zealand, as shown in the map in figure 1. Australia's Bureau of Meteorology has operated an EnviMeS MAX-DOAS instrument (similar instruments demonstrated good performance at the CINDI-2 MAX-DOAS intercomparison, see Kreher et al. (2019)) on a laboratory roof at its training facility
at Broadmeadows (37.690S, 144.947E, 110 m amsl) since December 2016. This location is close to some significant pollution sources including factories and major roadways. MAX-DOAS measurements of nitrogen dioxide and nitrous acid at the Broadmeadows site have been reported in Ryan et al. (2018).

Lauder is located in Central Otago, New Zealand (45.038S, 169.684E, 370 amsl), surrounded by irrigated farmland, ringed by distant mountain ranges and lying approximately 30 km north-east of the nearest large town, Alexandra. An EnviMeS
MAX-DOAS has been operational at Lauder since November 2016 allowing a significant period of overlap between the Lauder and Melbourne timeseries'. The NIWA EnviMeS MAX-DOAS demonstrated good performance at the CINDI-2 international comparison campaign held in the Netherlands in 2016 (Kreher et al., 2019).

Both Broadmeadows and Lauder have regular co-located meteorological, aerosol, radiation and and trace gas measurements; the Lauder site is part of numerous international atmospheric monitoring networks (De Mazière et al., 2018; Pollard et al.,





2017; Tradowsky et al., 2018). In addition, formaldehyde vertical columns measured at Lauder using fourier-transform infrared (FTIR) spectroscopy (Vigouroux et al., 2018) are available for comparison with the MAX-DOAS measurements.

The paper is structured as follows: the methodology section presents the MAX-DOAS and FTIR HCHO retrieval approach used in this work. Section 3.1 presents the MAX-DOAS HCHO timeseries from Lauder and Broadmeadows, briefly discussing potential HCHO sources at each location. Section 3.2 compares the FTIR and MAX-DOAS HCHO results from Lauder and
finally section 3.3 presents a comparison study for TROPOMI HCHO vs MAX-DOAS at both Lauder and Broadmeadows.

## 2 Methodology

### 2.1 MAX-DOAS measurements

MAX-DOAS measurements at Broadmeadows were made with a 2D EnviMeS instrument pointing to a fixed azimuth direction of 208°. The measurement, completed over 12 minutes, consisted of the elevation angles 90°, 30°, 20°, 10°, 5°, 3°, 2° and
1° as described in Ryan et al. (2018). At Lauder, a 1D EnviMeS instrument was used pointed at a fixed azimuth of 30° and the elevation angles used were 90°, 40°, 20°, 10°, 5°, 3° and 2°, and 1°. Dark current and offset corrections were made for each dataset using calibration spectra collected nightly, while initial wavelength and lineshape calibrations were facilitated by laboratory-measured mercury emission lamp spectra.

### 2.2 MAX-DOAS spectral analysis

The MAX-DOAS data analysis process consists of two parts: calculation of differential slant column densities (dSCDs) from the raw spectra and an inversion algorithm to retrieve vertical trace gas profiles from the dSCD information. The spectral retrieval was done in QDOAS (http://uv-vis.aeronomie.be/software/QDOAS/). Cross sections used in the analysis were $NO_2$ at 220 and 298 K (Vandaele et al., 1998), $O_4$ at 298 K (Thalman and Volkamer, 2013), $O_3$ at 223 and 243 K (Serdyuchenko et al., 2014), HCHO at 297 K (Meller and Moortgat, 2000), BrO at 223 K (Fleischmann et al., 2004), HONO at 298 K (Stutz
et al., 2000) and a Ring cross section at 250 K (Grainger and Ring, 1962). All cross sections were pre-convolved with the line shape of the instrument and 5th order polynomial and second order offset terms were also included in QDOAS.

Differential slant columns (dSCDs) of $O_4$, used in MAX-DOAS aerosol retrievals, were determined using the wavelength range 338-370 as in Ryan et al. (e.g. 2018); Kreher et al. (e.g. 2019). A simple sensitivity study was run to determine the appropriate wavelength range for formaldehyde retrieval given that two wavelength ranges are common in previous papers:
324.5-359 nm and 336-359 nm. Formaldehyde absorption bands for formaldehyde are in theory measurable by the MAX-DOAS UV spectrometers used in this work down to 300 nm. Published research to date, however, tends to avoid fitting below 320 nm however due to strong ozone absorption. Retrieval strategies in other work use a fitting range of 336-359 nm (e.g. Kreher et al., 2019; Heckel et al., 2005; Pinardi et al., 2013; Vigouroux et al., 2009) encompassing the three highest UV HCHO absorption features. Here a simple sensitivity study was run to determine if any benefit can be derived from additional
absorption bands in the extended range (e.g. Chan et al., 2019; Johansson et al., 2009; Wang et al., 2017b; Franco et al., 2015).

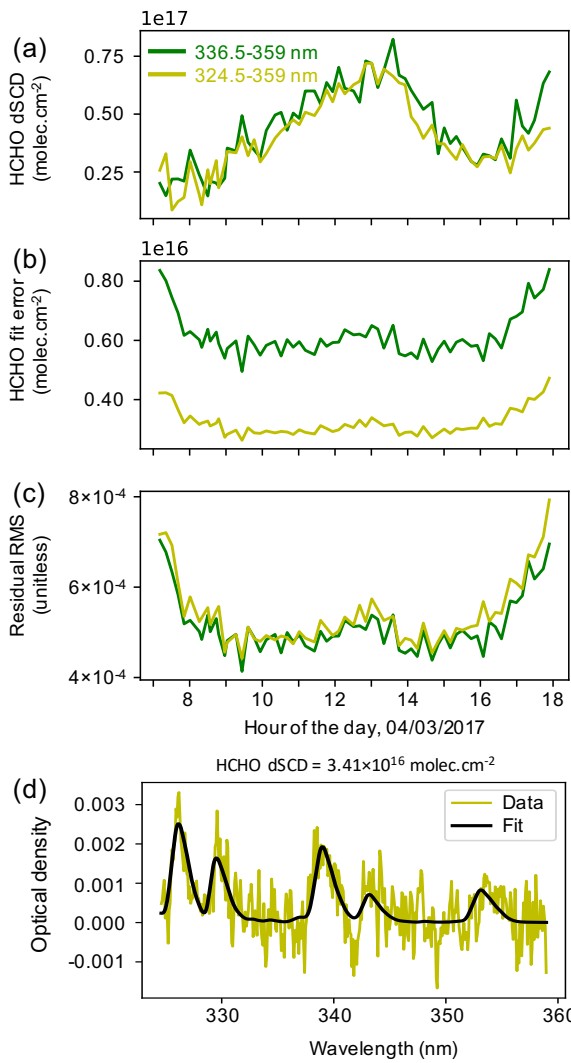

**Figure 2.** (a)-(c) Results of sensitivity tests to determine the appropriate fitting range for formaldehyde, using $3°$ elevation angle data from 04/03/2017 at Broadmeadows, showing dSCD, dSCD fit error and residual RMS respectively. (d) Example $3°$ elevation angle HCHO DOAS fit from 11 am local time on the same day.

Data for this test was chosen from a clear sky Autumn day at Broadmeadows with maximum HCHO dSCDs $\approx 7.5 \times 10^{16}$ molec.cm$^{-2}$ at $3°$ elevation angle. As shown in figure 2, the change of wavelength range had minimal impact on the residual RMS throughout the day, and minimal impact on the magnitude of the HCHO dSCD. However despite minimal change in dSCD, the fit error was greatly reduced by using the longer wavelength range, therefore the 324.5-359 nm range was adopted

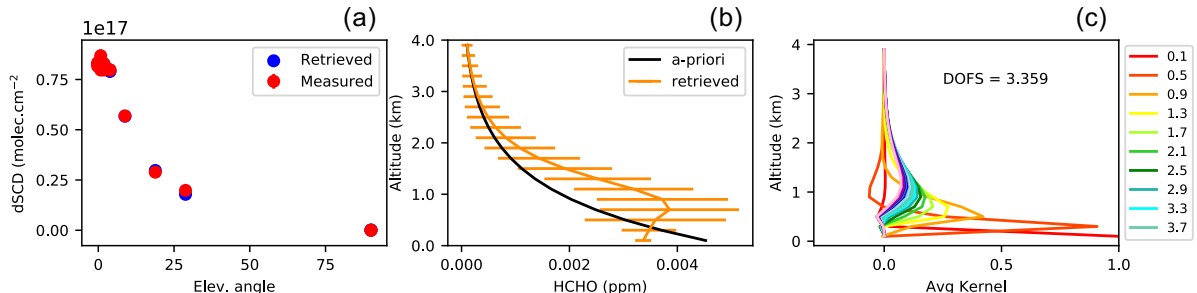

**Figure 3.** Example HEIPRO retrieval from Broadmeadows on 13/12/2017 at 14:00 local time. (a) shows model-measurement dSCD comparison, (b) shows the retrieved and a priori profiles, (c) the averaging kernels for this retrieval.

as the standard employed for fitting formaldehyde. An example HCHO DOAS fit is shown figure 2(d) demonstrating the convincing retrieval of formaldehyde dSCDs using the extended range.

## 2.3 MAX-DOAS profile retrievals

Formaldehyde vertical columns and profiles from Broadmeadows and Lauder were retrieved from dSCDs using the HEIPRO inversion algorithm (Frieß et al., 2006). HEIPRO has previously been used for $NO_2$ and HONO gas profile retrievals at Broadmeadows (Ryan et al., 2018). In an initial step, aerosol profiles were determined from dSCDs of the $O_4$ dimer. These were used as input information on the light path for calculating air mass factors and HCHO vertical column density (VCD) in the second retrieval step. Vertical profiles were retrieved on a 20 layered grid with 200 m resolution from 0-4 km, aerosol retrievals were calculated at 360.8 nm and HCHO retrievals at 338.9 nm. A priori profiles used in the inversion were chosen to be exponentially decreasing functions of altitude, characterised by a set surface mixing ratio and scale height, which were 0.5 ppb and 1 km respectively for formaldehyde. According to sensitivity tests undertaken for aerosol retrievals and nitrous acid at Broadmeadows in Ryan et al. (2018), the uncertainty of the trace gas vertical columns as a result of uncertainty in trace gas and aerosol a priori parameters is less than 10 %. HEIPRO was run in 15 minute intervals ensuring that each measurement set contained a full set of elevation angles. MAX-DOAS retrievals were filtered for results with less than one independent piece of information and for the presence of clouds. At Broadmeadows this was determined using an empirical algorithm based on colour indices (e.g. Gielen et al., 2014; Wagner et al., 2014, 2016), also described in Ryan et al. (2018), and at Lauder using the SkyNet AOD flag which is calculated using the method outlined in Khatri and Takamura (2009).

An example MAX-DOAS HCHO retrieval from HEIPRO is shown in figure 3, including model-measurement comparison, retrieved and a priori profile and averaging kernels. These example averaging kernels at Broadmeadows show highest sensitivity at the surface, and 3.4 degrees of freedom for signal (DoFs). The Lauder retrievals consistently had reduced surface sensitivity and lower DoFs compared to Melbourne, which is likely related to lower amounts of formaldehyde at Lauder and the fact that $2°$ is the lowest possible elevation angle for MAX-DOAS at Lauder due to proximate mountain ranges. Across the whole





measurement period, the average DoFs was $2.25 \pm 0.34$ ($1\sigma$) at Broadmeadows and $1.27 \pm 0.11$ ($1\sigma$) at Lauder. Detection limits for the MAX-DOAS vertical column densities at Lauder and Broadmeadows have been estimated using the method outlined in Peters et al. (2012):

$$DL_{VCD} = \frac{2 \times R_{avg}}{XS_{max} \times A - 1} \qquad (1)$$

where $R_{avg}$ is the average residual RMS, $XS_{max}$ is the maximum value of the cross section ($1.32 \times 10^{-19}$ for HCHO) and A is the airmass factor taken here as 15 for low elevation angles. $R_{avg}$ was $4.5 \times 10^{-4}$ at Broadmeadows giving $DL_{VCD}(HCHO)$ as $4.9 \times 10^{14}$ molecules.cm$^{-2}$. The average residual RMS was lower at Lauder, $2.9 \times 10^{-4}$, giving a calculated detection limit of $3.2 \times 10^{14}$ molecules.cm$^{-2}$. Over the whole measurement period, the average vertical column was $3.13 \pm 0.61 \times 10^{15}$ molec.cm$^{-2}$ at Lauder and $5.40 \pm 1.59 \times 10^{15}$ molec.cm$^{-2}$ at Broadmeadows, meaning that HCHO VCDs were generally above the detection limit but measurements at Lauder were closer to the detection limit than at Broadmeadows.

## 2.4 FTIR retrieval

Solar FTIR measurements have been made since the early 1990s at Lauder as part of the Network for Detection of Atmospheric Composition Change (NDACC, (Jones et al., 1994; De Mazière et al., 2018). Measurements are made on all possible clear sky days, throughout the day, using Bruker high resolution ($0.0035$cm$^{-1}$) spectrometers (www.bruker.com).

Initial retrievals of HCHO from the Lauder 1992-2005 FTIR dataset are described in detail in Jones et al. (2009). The HCHO retrieval strategy (under the auspices of the NDACC infrared working group) was harmonized across the network as detailed in Vigouroux et al. (2018). Lauder spectra HCHO reprocessing was part of this harmonization activity and is the retrieval strategy used to provide HCHO data in this study. The same HCHO dataset is also used in a TROPOMI comparison study comprising of globally distributed ground based FTIR measurements (Vigouroux et al., 2020). These studies show that HCHO abundances over Lauder exhibit a seasonal cycle peaking in the summer (DJF).

Pertinent to this study, and paraphrasing details in Vigouroux et al. (2018), the Lauder FTIR retrievals are performed on a 48-layer atmosphere (0.37-100km) of which 15 layers are between 0.37 and 10km. The retrievals use a static a priori originating from WACCM_v4 climate-chemistry model simulations (Garcia et al., 2007) and the retrievals are constrained using Tikhonov regularization (L1, alpha = 100). Combined with a measurement signal to noise ratio of 400, the retrieval strategy has sensitivity over the altitude range 0.37-26km with an average total column DoFs of $1.4 \pm 0.2$ ($1\sigma$). The highest sensitivity is in the upper troposphere peaking at 8km with a full width maximum of 16-18km. This differs from the MAX-DOAS measurements which has maximum sensitivity in the boundary layer. An example Lauder FTIR formaldehyde retrieval from 8th January 2018 is shown in figure 4. Attributed uncertainty analysis of the total column measurement gives an estimate of $\approx 2\%$ and $\approx 12\%$ for random and systematic error respectively. The systematic error is dominated by spectroscopic line strength uncertainty whilst the major component of the random error is measurement noise.





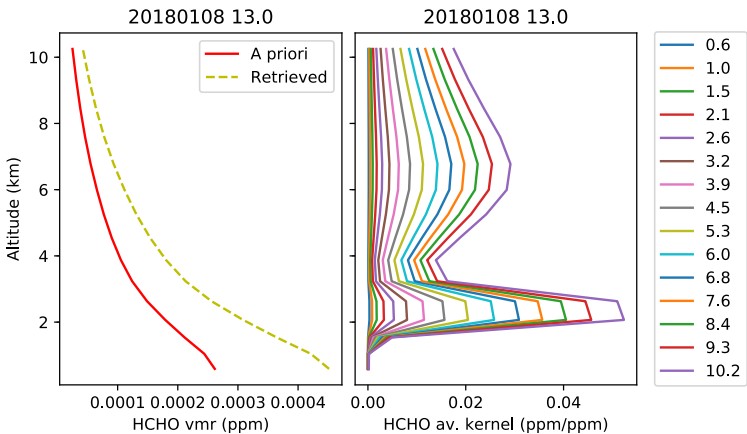

**Figure 4.** Example FTIR profile retrieval from 1 pm local time on 8th January 2018 at Lauder. The left hand panel shows a priori and retrieved HCHO profiles in the lowest 10 km, the right hand panel shows the corresponding averaging kernels.

## 2.5 Satellite details

The TROPOspheric Monitoring Instrument (TROPOMI) is a nadir viewing imaging spectrometer aboard the European Space Agency's Copernicus Sentinel 5 Precursor (S5P) satellite. S5P, launched in October 2017, is a low (afternoon) polar orbit ($\approx 824$ km) mission providing daily global coverage for a range of UV, visible and infrared absorbing trace gases (Veefkind et al., 2012). The S5P overpass time is 13:30 (local time) and the spatial resolution of TROPOMI is $3.6 \times 7.2$ km (before 6 August 2019) and $3.6 \times 5.6$ km (after 6 August 2019).

Formaldehyde slant column densities are retrieved from analysis of absorption features over the wavelength range 328.5 - 359 nm. The SCDs are converted to vertical columns using airmass factors calculated at 340 nm using HCHO a priori vertical profiles simulated by the TM5-MP global chemistry-transport model as described in De Smedt et al. (2018).

For this study, TROPOMI data was regridded to $0.1 \times 0.1°$, meaning that average columns approximately 10 km either side of the MAX-DOAS measurement site were used. The recommended quality control (QC) filtering was applied, excluding retrieved values where the QC flag was less than 0.5 (on a scale of 0-1), and cloudy scenes as determined by the satellite results were also excluded from the comparisons. Given the satellite overpass was around 13:30 local time, MAX-DOAS results between 13:00 and 14:00 were averaged for the comparisons.

The Ozone Monitoring Instrument (OMI) is also a UV/Vis nadir viewing spectrometer providing near global daily coverage, housed on the National Aeronautics and Space Adminsitration's Earth Observing System Aura satellite (Levelt et al., 2006). The spatial resolution of OMI is $13 \times 24$ km and the overpass time is also around 13:30 local time. Formaldehyde slant columns are retrieved from OMI using a wavelength range of 327.5 - 356.5 nm (González Abad et al., 2015) are used along with GEOS-Chem simulated a priori profiles to calculate HCHO vertical columns (Bey et al., 2001). For comparison





with the Broadmeadows MAX-DOAS dataset, OMI HCHO columns were regridded to $0.25 \times 0.25°$, meaning that columns approximately 25 km either side of the measurement site were used, and as with TROPOMI, cloudy scenes were excluded from the comparison.

## 3  Results and Discussion

### 3.1  Lauder vs Melbourne HCHO

The timeseries' of monthly formaldehyde vertical columns from Broadmeadows and Lauder MAX-DOAS measurements are presented in figure 5(a). Following the example of Jones et al. (2009), the seasonal cycle of was fitted with a cosine function described by the following equation:

$$C(t) = a_0 + a_1 t + a_2 \cos K (t - \phi) \tag{2}$$

where $C(t)$ is the formaldehyde vertical column as a function of time (in units of days since 1st January 2016), $\phi$ is the phase term with units of day of the year and $K = 2\pi/365$. Also fitted in the linear regression are $a_2$ (amplitude of the seasonal cycle), $a_0$ (the initial mean column amount) and $a_1$ (the magnitude of the linear trend in HCHO over time). At Lauder, the mean HCHO VCD was $2.5 \times 10^{15}$ molecules.cm$^{-2}$ and the amplitude of the fitted seasonal cycle was $6.9 \times 10^{14}$ molecules.cm$^{-2}$ while at Broadmeadows the average HCHO VCD was $5.4 \times 10^{15}$ molecules.cm$^{-2}$ with a seasonal cycle amplitude of $2.0 \times 10^{15}$ molecules.cm$^{-2}$. The HCHO seasonal cycle from Lauder MAX-DOAS measurements is consistent with that found from FTIR measurements at Lauder from July 2002-July 2017 (Vigouroux et al., 2018). The fact that both the magnitude of the HCHO VCDs and amplitude of the seasonal cycle are much smaller at Lauder than Broadmeadows could be due to higher anthropogenic VOC precursors as Melbourne is a large city, or could be due to higher biogenic VOC emissions from forests surrounding Melbourne.

The seasonal cycle of formaldehyde shows a distinct austral summer peak in both locations. This would be expected from biogenic production of formaldehyde, for example from isoprene, which depends strongly on temperature (Duncan et al., 2009; Palmer et al., 2006; Zhu et al., 2014). The phase of the cosine fit in each location is 31 days indicating that the HCHO seasonal cycle peaks at the end of January. This is also consistent with the results for Lauder in Vigouroux et al. (2018) and suggests that the same background mechanisms may be responsible for summer-time HCHO production at Lauder and Broadmeadows.

Polar bivariate plots showing the relationship between formaldehyde and wind direction and speed at Broadmeadows and Lauder are shown in figure 5(b) and (c) respectively. At Broadmeadows, HCHO concentrations are highest with wind from the northern and eastern sectors, aligning with the direction of rural and densely forested regions, suggesting an important role for biogenic HCHO sources in this location. At Lauder, maximum column amounts correspond with moderate wind speeds from the east. Over the course of the MAX-DOAS dataset, the wind came from this direction less than 10 % of the time, the same key source directions including the strong 'easterly maximum' are observed in polar bivariate plots of the 2001-2019 FTIR dataset (not shown). There is a large variation in vegetation types across New Zealand's South Island, including temperate



rainforest in the west, dryland agricultural in the Central Otago region and intensive irrigated pasture in much of the east, south and south east, which might be expected to produce different volatile organic emissions and formaldehyde amounts. Given that the lifetime of formaldehyde is on the order of hours, transport on the order of a hundred kilometers is possible meaning the different source directions can reasonably be compared. On the available evidence it could be hypothesised that the agricultural eastern sector is a stronger source of formaldehyde to Lauder than the forested the west coast.

## 3.2 MAX-DOAS vs FTIR at Lauder

One previous study highlights a comparison between MAX-DOAS and FTIR formaldehyde columns, from the tropical Reunion Island (Vigouroux et al., 2009). In that paper, the comparison period was four months. In this work, co-located measurements over a period of 27 months are compared, from November 2016 to January 2019, allowing comparison of HCHO over two annual cycles. The comparison method used here has been adapted from Vigouroux et al. (2009) and Rodgers and Connor (2003). Partial column amounts have been compared in the lowest 4 km of the atmosphere, the region of expected formaldehyde production and the region of highest sensitivity for MAX-DOAS measurements. Because the FTIR instrument is less sensitive to the HCHO partial column in the lowest 4 km as evident from the averaging kernels in figures 3(a) and 4), the MAX-DOAS partial columns have been smoothed by the FTIR total averaging kernel using the method outlined in Vigouroux et al. (2009). As in Vigouroux et al. (2009), the equation for the smoothing is simplified by the fact that the same a priori profile was used to retrieve MAX-DOAS and FTIR profiles, allowing the smoothed DOAS column to be given by:

$$C_{DOAS,smooth} = C_a + \mathbf{A_F}(\mathbf{x_D} - \mathbf{x_a}) \tag{3}$$

where $\mathbf{A_F}$ is the FTIR total column averaging kernel matrix (from 0-4 km), which has units of VMR/VMR, $C_a$ is the common a priori column amount, $\mathbf{x_D}$ is the original retrieved MAX-DOAS profile, $\mathbf{x_a}$ is the common a priori profile and $C_{DOAS,smooth}$ is the smoothed MAX-DOAS column amount. Only columns between 8 am and 6 pm local time contributed to the monthly averages examined here.

The timeseries of monthly averaged results is presented in figure 6(a), showing that both measurements capture the same broad seasonal cycle at Lauder and that monthly average columns for both measurements were clearly above the calculated MAX-DOAS detection limit. The month-to-month variation in formaldehyde is in especially good temporal agreement for summer (DJF) 2017-18, while the summer increase is less clearly captured by the FTIR in summer 2016-17 or 2018-19. There is a clear offset between the MAX-DOAS and FTIR columns with the FTIR consistently lower across the comparison period. Comparing the measurements by linear (Deming) regression, (incorporating errors in both the x and y ordinates), the offset is found to be $1.02 \times 10^{15}$ molecules.cm$^{-2}$ and nearly constant as indicated by the regression slope (0.91, see figure 6(b)). The timeseries also shows that smoothing the DOAS partial columns brought them more into line with the FTIR columns, especially in the peak months (November-March). The R$^2$ value of 0.81 (n=27) for the regression in figure 6(b) highlights the strong temporal agreement. Considering daily averages, a slope of 0.64 and R$^2$ of 0.46 (n=810) were found, while for weekly

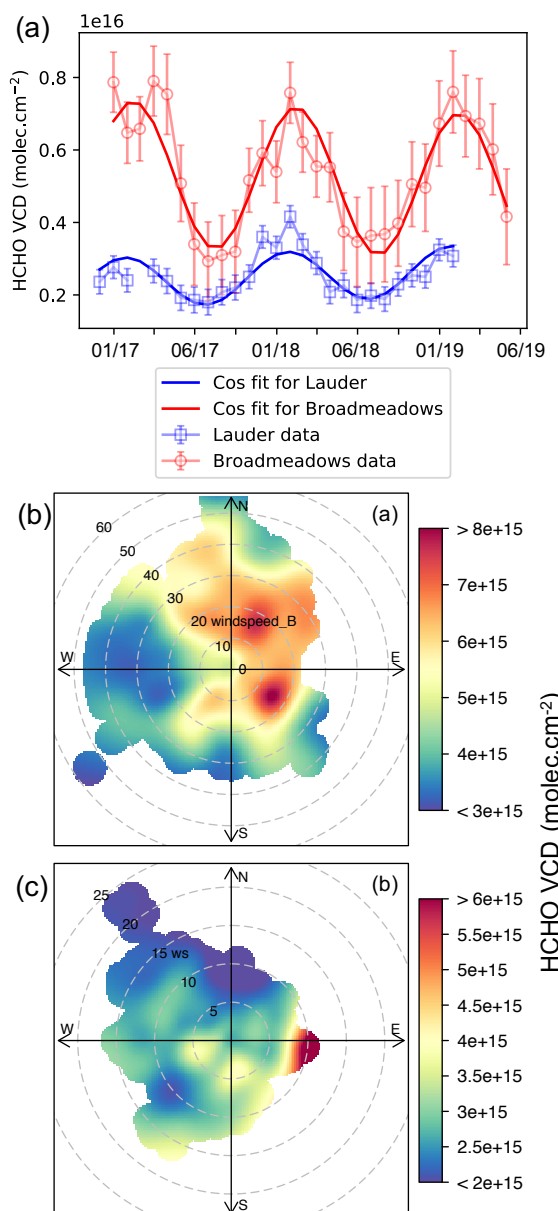

**Figure 5.** (a) Timeseries of monthly averaged MAX-DOAS formaldehyde VCDs at Broadmeadows (red points) and Lauder (blue squares). Cosine fits to each dataset are also plotted (solid lines), representing the seasonal cycle of HCHO at each location. (b) Polar bivariate plot showing the relationship between HCHO VCD and windspeed (in km/h) and direction at Broadmeadows across the measurement period. (c) Same as (b) but for Lauder.





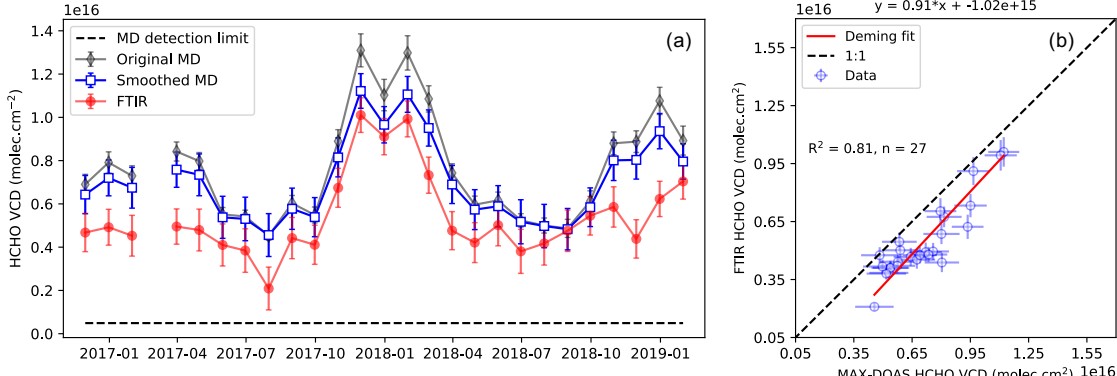

**Figure 6.** (a) Timeseries comparison of monthly averaged FTIR and MAX-DOAS (original and smoothed with FTIR averaging kernels) HCHO vertical columns from November 2016 to January 2019 at Lauder. (b) Scatter plot comparison of FTIR with MAX-DOAS smoothed HCHO columns.

averages the slope of the Deming regression was 0.71 with $R^2 = 0.59$ (n=116). The weekly and daily average timeseries and scatter plots are shown in figure A1 in Appendix A.

The percentage differences between the MAX-DOAS and FTIR (mean $\pm$ standard deviation) were $63 \pm 36\%$ for the original MAX-DOAS columns and $53 \pm 36\%$ when comparing the smoothed MAX-DOAS columns. As expected given the offset between FTIR and MAX-DOAS column amounts, the differences were on average greater than the combined error on the differences of the columns, calculated according to $E_{TC} = \sqrt{E_{MAX-DOAS}^2 + E_{FTIR}^2}$, where $E_{FTIR}$ and $E_{MAX-DOAS}$ are the combined random and systematic errors of each instrument. For the whole timeseries the mean combined error on the

differences was $16.7 \pm 4.9\%$ (mean $\pm 1\sigma$). The differences between FTIR and MAX-DOAS columns here are greater than for the results found in the Reunion Island comparison (Vigouroux et al., 2009), where no significant offset between measurements was found. In contrast to their study, here the smoothing was found to improve the mean difference between the columns. The greater mean difference and standard deviations of the differences at Lauder compared to Vigouroux et al. (2009) likely reflect the much longer comparison period, incorporating variations across a much wider range of atmospheric conditions, and the fact

that here only the altitude range 0-4 km is examined rather than 0-10 km as in Vigouroux et al. (2009). In addition, differences in site characteristics may play a role in the greater offset observed at Lauder. Reunion Island being a coastal site is likely to be measuring marine background formaldehyde, as indicated by the fact that the 2007 measurements in Vigouroux et al. (2009) rarely not exceeded $7.7 \times 10^{15}$ molecules.cm$^{-2}$, with little local surface HCHO production. In comparison, the mean smoothed DOAS column across the 27 month comparison period was $7.7 \times 10^{15}$ molecules.cm$^{-2}$, suggesting greater local production

which will occur at the surface, where the MAX-DOAS sensitivity is greatest and the FTIR least sensitive.

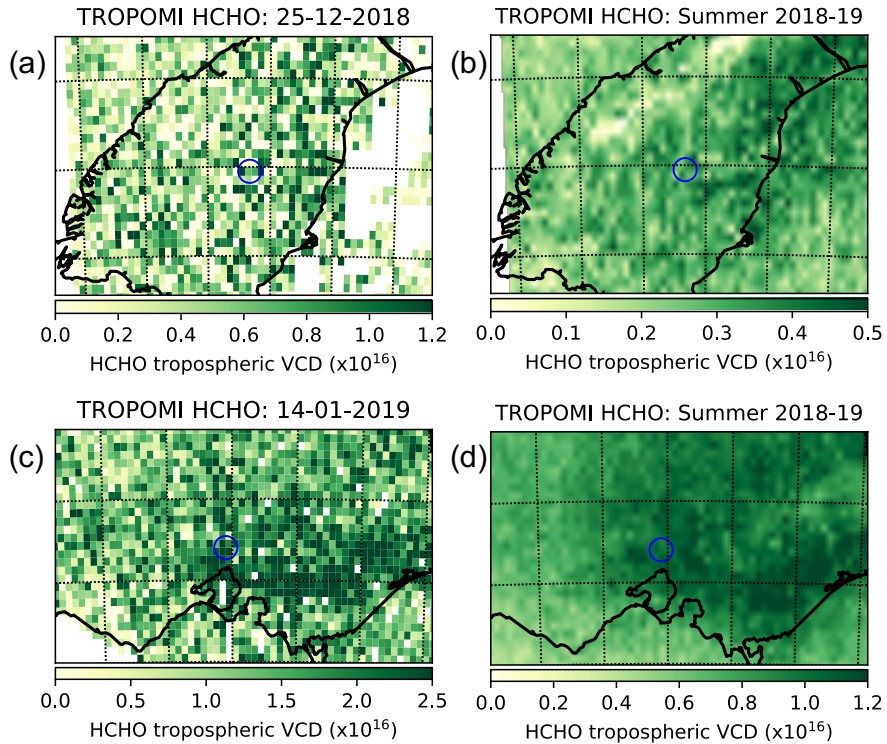

**Figure 7.** Examples of TROPOMI $0.1 \times 0.1°$ gridded output. The blue circles indicate the MAX-DOAS measurement locations. The grid lines drawn are at $1°$ latitude and longitude intervals. (a) Example retrieval from the South Island of New Zealand, 24th of December 2018. (b) All New Zealand retrievals from December 2018-February 2019 averaged. (c) Example from Victoria, 14th January 2019, and (d) all Victorian retrievals from December 2018-February 2019 averaged. Note that the colour bar scale is different for each plot to emphasise the spatial patterns.

## 3.3 MAX-DOAS vs TROPOMI

In this section, MAX-DOAS formaldehyde columns are compared with satellite results. Firstly, Lauder HCHO MAX-DOAS columns are examined alongside results from TROPOMI. Following the example of MAX-DOAS vs satellite formaldehyde comparisons in Hoque et al. (2018b) and De Smedt et al. (2015), vertical columns are compared rather than profiles.

TROPOMI reports an uncertainty on the column amount, however it was found that this uncertainty was highly correlated with the magnitude of the column amount. Therefore, we estimated the uncertainty on the satellite column retrievals from the number of retrievals contributing to the averaged column in the $0.1 \times 0.1°$ grid box (number per cell, $N_{pc}$) and the standard deviation of those retrievals ($S_T$): $E_{sat} = \frac{S_T}{\sqrt{N_{pc}}}$. More measurements were available from TROPOMI over Broadmeadows than at Lauder, with the average $N_{pc}$ across the comparison period and considering TROPOMI pixels $1°$ either side of the

ground-based station, of 1.18 in New Zealand and 2.76 in Melbourne. Final compared results filtered out pixels with $N_{pc} < 1$,





giving a used average $N_{pc}$ of 1.84 for Lauder and 2.94 for Broadmeadows. The discrepancy in $N_{pc}$ could be due to more cloud over New Zealand than Victoria, or because HCHO columns over Lauder are low enough to be approaching the detection limit. TROPOMI results showed greater spatial variation over New Zealand than Victoria, as illustrated in the example map in figure 7(a). This is reflected in the standard deviation ($S_T$) of HCHO retrievals contributing to the Lauder and Broadmeadows average

TROPOMI columns; the mean $\pm S_T$ was $1.66 \times 10^{15} \pm 1.50 \times 10^{15}$ molec.cm$^{-2}$ and $7.53 \times 10^{15} \pm 1.10 \times 10^{15}$ molec.cm$^{-2}$ for Lauder and Broadmeadows respectively. Overall, these factors combined to give a high mean percentage variancce for Lauder TROPOMI columns of 129 %, while for Broadmeadows it was only 9.7%.

  Nevertheless, the average summer (DJF) 2018-19 TROPOMI retrieval map for the central New Zealand South Island shown in figure 7(b), supports the conclusion (from the MAX-DOAS measurements) that highest formaldehyde amounts are in the

agricultural eastern parts of the island. There are no standout HCHO hot spots in the thickly forested west coast or south-western Fiordland regions. The New Zealand Alps are highlighted in this figure by the lack of formaldehyde, possibly due to minimal vegetation in this region and because the satellite retrieval will not work over areas of high albedo (i.e. snow). The inference that formaldehyde is close to background levels is supported by the fact that average summer column amounts over the Tasman Sea and Pacific Ocean off the coast of the South Island appear similar to those over land. In comparison,

the average summer 2018-19 map from Victoria highlights some clear features especially high formaldehyde levels over the densely forested regions in the east of the state. The irrigated agricultural land north of Melbourne stands out compared to the drier grazing country in the west and north-west; these areas highlighted by TROPOMI correspond to the directions of highest measured HCHO at Broadmeadows in figure 5(b).

  Formaldehyde columns from TROPOMI and MAX-DOAS at Broadmeadows and Lauder were compared over the course

of 18 months (May 2018-November 2019). For the comparison, TROPOMI results (columns and associated a priori profiles and averaging kernels) were averaged $0.1°$ either side of the Broadmeadows and Lauder MAX-DOAS locations. MAX-DOAS columns (along with averaging kernels) were averaged between 13:00-14:00 local time, around the time of the TROPOMI overpass. TROPOMI vertical profiles are not available for download and hence, to accurately compare tropospheric columns across the same altitude range, the MAX-DOAS retrievals for this comparison were run to 10 km rather than 4 km as in the

FTIR-MAX-DOAS comparison in section 3.2.

  For direct comparison of TROPOMI and MAX-DOAS formaldehyde vertical columns, accounting for the different instrumental a priori profiles and vertical sensitivities, the method outlined in Vigouroux et al. (2020) for comparing TROPOMI with FTIR was adapted. Firstly, to account for the fact that the two retrieval methods usec different a priori profiles, the following equation was used to produce an adjusted MAX-DOAS profile $\mathbf{x'_D}$:

$$\mathbf{x'_D} = \mathbf{x_D} + (\mathbf{A_M} - \mathbf{I})(\mathbf{x_{D,a}} - \mathbf{x_{T,a}}) \tag{4}$$

  where $\mathbf{x_D}$ is the original MAX-DOAS profile, $\mathbf{A_M}$ is the MAX-DOAS averaging kernel matrix, $\mathbf{I}$ is the identity matrix, $\mathbf{x_{D,a}}$ is the MAX-DOAS a priori profile and $\mathbf{x_{T,a}}$ is the TROPOMI a priori profile expressed on the MAX-DOAS altitude grid. The integrated adjusted column gave an adjusted MAX-DOAS HCHO tropospheric column, which was then smoothed using

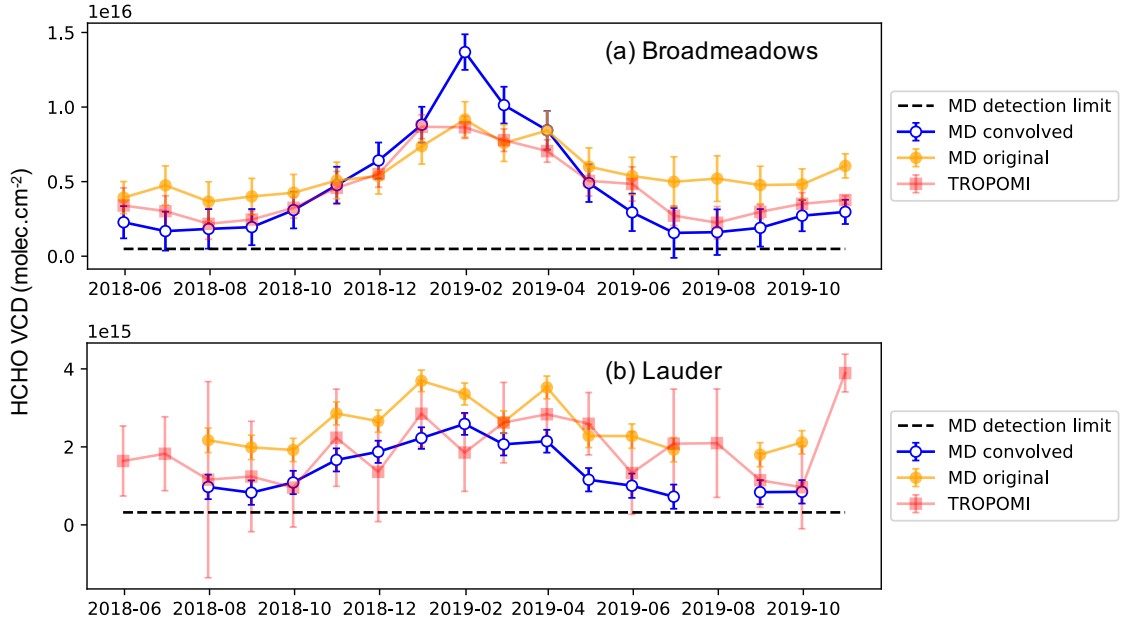

**Figure 8.** Timeseries comparison of monthly mean formaldehyde columns from TROPOMI and the MAX-DOAS between May 2018 and November 2019 at (a) Broadmeadows and (b) Lauder. Both the original (orange) and convolved (blue and white, i.e. accounting for different retrieval a priori and sensitivities) MAX-DOAS columns are shown.

the TROPOMI averaging kernels (expressed on the MAX-DOAS altitude grid) using the same method as for smoothing the
FTIR columns in section 3.2 (Rodgers and Connor, 2003):

$$C_{D,smooth} = C_{T,a} + \mathbf{a_T}(\mathbf{x'_D} - \mathbf{x_{T,a}}) \tag{5}$$

where $C_{D,smooth}$ is the smoothed MAX-DOAS tropospheric column, $C_{T,a}$ is the TROPOMI a priori tropospheric column and $\mathbf{a_T}$ is the TROPOMI column total averaging kernel.

The monthly average timeseries of HCHO tropospheric columns at Broadmeadows measured by MAX-DOAS and TROPOMI
is shown in figure 8(a). The seasonal variation of formaldehyde with its strong summer peak is clearly captured by TROPOMI, with all MAX-DOAS and TROPOMI data points above the calculated MAX-DOAS detection limit. The original MAX-DOAS retrieved columns agree well the magnitude of the TROPOMI observations between October 2018 and June 2019, including over the summer peak, but are greater than TROPOMI outside these months. The MAX-DOAS columns adjusted for a priori differences and convolved with TROPOMI averaging kernels agree well with TROPOMI, within uncertainty, for all months
except the height of the summer peak in January-February 2019. This discrepancy during times of peak HCHO production in the boundary layer highlights the much greater sensitivity of the MAX-DOAS to the lower atmosphere than TROPOMI. On

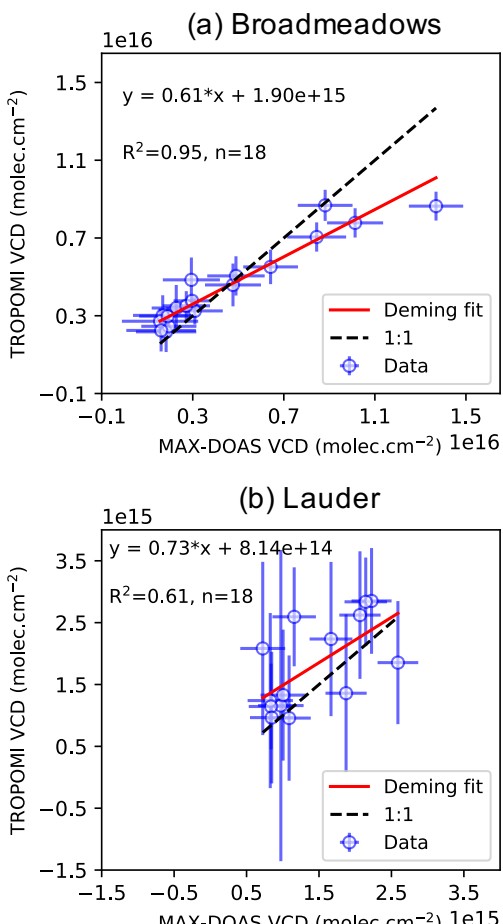

**Figure 9.** Scatter plot and Deming regression results for the comparison of monthly mean formaldehyde columns from TROPOMI and the MAX-DOAS (convolved with TROPOMI averaging kernels) between May 2018 and November 2019 at (a) Broadmeadows and (b) Lauder.

average the difference between TROPOMI and the smoothed MAX-DOAS HCHO tropospheric columns at Broadmeadows was $21\% \pm 34\%$.

Figure 8(b) shows the same as figure 8(a) but for Lauder. As with Broadmeadows, the broad seasonal variation is captured

by TROPOMI and all data points are above the calculated MAX-DOAS detection limit although TROPOMI errorbars are greater than at Broadmeadows, often extending below the MAX-DOAS detection limit, due to the lower number of available TROPOMI retrievals over Lauder. The convolved MAX-DOAS HCHO columns compare well within error for a majority of months. As with Broadmeadows, TROPOMI columns were greater on average than MAX-DOAS, by an average of $38 \pm 59\%$, although the difference was much less up to April 2019 ($12 \pm 30\%$).





The agreement between TROPOMI and MAX-DOAS is further examined using linear Deming regression analysis in figure 9. For Lauder, figure 9(b) shows the monthly average scatter plot with overall regression slope of 0.73 and $R^2 = 0.61$ (n=18). The majority of data points lie are within error of the 1:1 line. The regression values for the daily measurements at Lauder were slope = 0.40 and $R^2 = 0.22$ (n=510), while weekly averages gave a slope of 0.66 and $R^2$ of 0.45 (n=73). At Broadmeadows, data points lie along the 1:1 line within error except for the highest two values, which are January and February 2019 as

highlighted in the timeseries, giving a regression slope of 0.61. The very strong temporal agreement is highlighted by an $R^2$ of 0.95 (n=18). Considering the individual daily measurements at Broadmeadows, the slope of the regression was 0.77 with $R^2 = 0.69$ (n=506) and for weekly averages the slope was 0.66 with $R^2 = 0.89$ (n=73) (plots for Lauder and Broadmeadows daily measurements and weekly averages are shown in figures A2 and A3 in Appendix A.

        The success of this comparison study for formaldehyde with TROPOMI, especially at Broadmeadows, is highlighted by

a comparison (2017-2019) at the same Broadmeadows location between OMI and the MAX-DOAS. As shown in figure A4, OMI does not clearly capture any of the seasonal formaldehyde variation in Melbourne, and as such fails to replicate the MAX-DOAS values. The errorbars shown in this figure are the quoted uncertainty on the OMI columns, and represent 67 % of the total column on average, perhaps due to the poorer resolution of OMI compared to TROPOMI, making observation of the seasonal cycle difficult in this data. Monthly OMI HCHO columns are on average 200 % higher than the MAX-DOAS (see table A1 in

Appendix A), far greater than any discrepancy reported in the literature for a MAX-DOAS vs satellite retrieval. One possibility for the disparity is inappropriate a priori profiles, which are calculated in OMI using chemical transport models. Knowing that formaldehyde production from isoprene and monoterpene emissions is poorly simulated in south-eastern Australia using standard VOC mechanisms (Emmerson et al., 2016, 2018), relying on these mechanisms to produce useful formaldehyde a priori profiles in this area may be problematic. Previous studies (e.g. De Smedt et al., 2015; Wang et al., 2017a) found that

agreement between OMI and MAX-DOAS measurements was found when using the MAX-DOAS a priori profiles to retrieve satellite columns; it would be interesting in future work to do the same for HCHO satellite-based retrievals over Australasia.

## 4   Conclusions

This paper presents comparison studies of MAX-DOAS formaldehyde measurements in two distinctly different environments: the remote Central Otago region in New Zealand and the suburban fringe area of Broadmeadows in Victoria. This work is the

first long-term comparison and validation study undertaken using MAX-DOAS measurements in the Southern Hemisphere.

        For MAX-DOAS measurements between December 2016 and November 2019, the mean formaldehyde column measured by the MAX-DOAS at Broadmeadows was $5.40 \pm 1.59 \times 10^{15}$ molec.cm$^{-2}$ compared to $2.50 \pm 0.61 \times 10^{14}$ molec.cm$^{-2}$ at Lauder. The amplitude of the seasonal cycle was also greater at Broadmeadows than Lauder, $2.0 \times 10^{15}$ molec.cm$^{-2}$ compared to $0.7 \times 10^{15}$ molec.cm$^{-2}$. The seasonal cycles at Lauder and Broadmeadows could be described by a periodic function peaking

at the end of January, i.e. at the height of the austral summer, consistent with biogenic temperature-dependent formaldehyde production.





At Lauder, 27 months of MAX-DOAS measurements were compared with FTIR formaldehyde partial columns between 0-4 km. Smoothing of the FTIR columns using the MAX-DOAS averaging kernels to resolve for the different vertical sensitivities was carried according to the methodology outlined in Rodgers and Connor (2003) and Vigouroux et al. (2009). The seasonal cycle of formaldehyde at Lauder, with pronounced summer peak, was clearly replicated by both sets of observations and the smoothed FTIR columns correlated more strongly than the original with the MAX-DOAS results. The timing of the HCHO seasonal cycle peak were very similar between Broadmeadows and Lauder suggesting similar HCHO sources, however the source strength at Lauder seems to be weaker with a lower seasonal cycle amplitude.

In the first TROPOMI - MAX-DOAS Southern Hemispheric comparison study, TROPOMI performed especially well compared to the Broadmeadows monthly average columns in terms of temporal variation and magnitude ($R^2 = 0.95$, slope = 0.81). This result is a significant improvement in both the comparison with OMI both at this location and in previous literature reports. Higher spatial variability and lower absolute amounts of HCHO made the comparison more difficult at Lauder, however, the linear regression analysis also indicated moderate temporal agreement in most months of the comparison ($R^2 = 0.61$, slope = 0.73).

Using maps of average TROPOMI HCHO retrievals this study also demonstrates the utility of the satellite product to identify hotspot regions of biogenic VOCs, which will be critical tool in addressing the current gap in understanding of isoprene and monoterpene chemistry in south-eastern Australia.

This TROPOMI comparison study, especially over Melbourne, raises many exciting possibilities for future work. This study shows the importance of long-term timeseries MAX-DOAS measurements for satellite validation, and could be contributed to international validation efforts. This study could also be extended to consider not only formaldehyde validation but also $NO_2$, HONO and glyoxal. This would begin to address the current Northern-Hemispheric bias in satellite validation studies. This work also shows the utility of the MAX-DOAS technique for studying formaldehyde in the VOC-hotspot of south-eastern Australia, and it would be interesting in future studies to deploy MAX-DOAS instruments into the forested areas highlighted in TROPOMI as large formaldehyde source regions. Moreover, this work has shown that improvements in satellite technology, culminating (at this point in time) in TROPOMI, mean that space-based HCHO measurements will also be of great benefit in constraining the temporal and spatial distribution of VOC emissions in this region. With such assurance, related tropospheric oxidation and ozone chemistry, with their associated air quality and climate implications, can be studied on a much grander scale.

*Data availability.* Data available on request from the authors





**Appendix A**

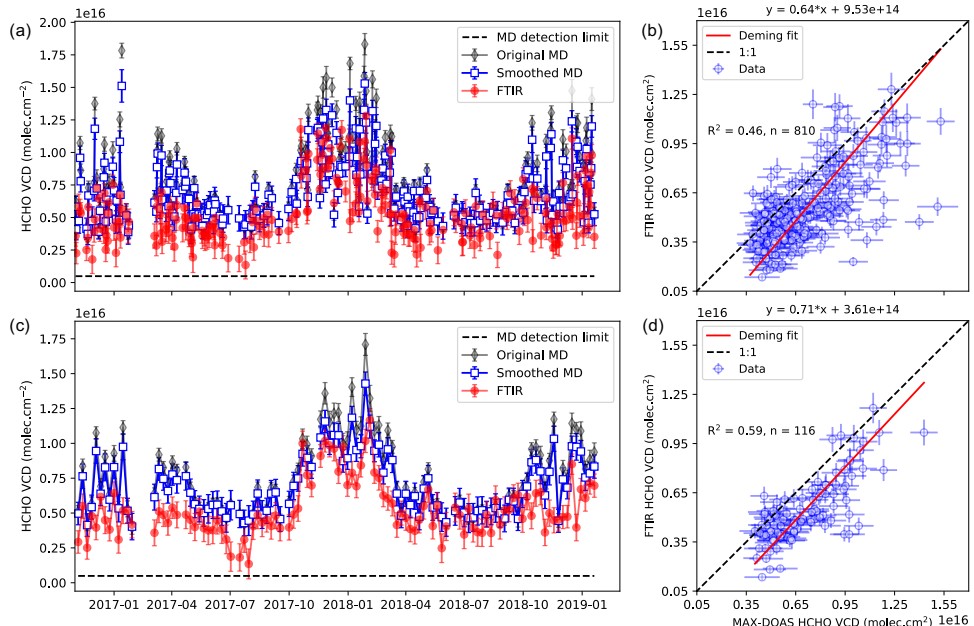

**Figure A1.** Results for the comparison of formaldehyde columns from FTIR and the MAX-DOAS (convolved with FTIR averaging kernels) at Lauder: (a) Daily average timeseries, (b) daily average scatter plot weekly averages, (c) weekly average timeseries and (d) weekly average scatter plot.



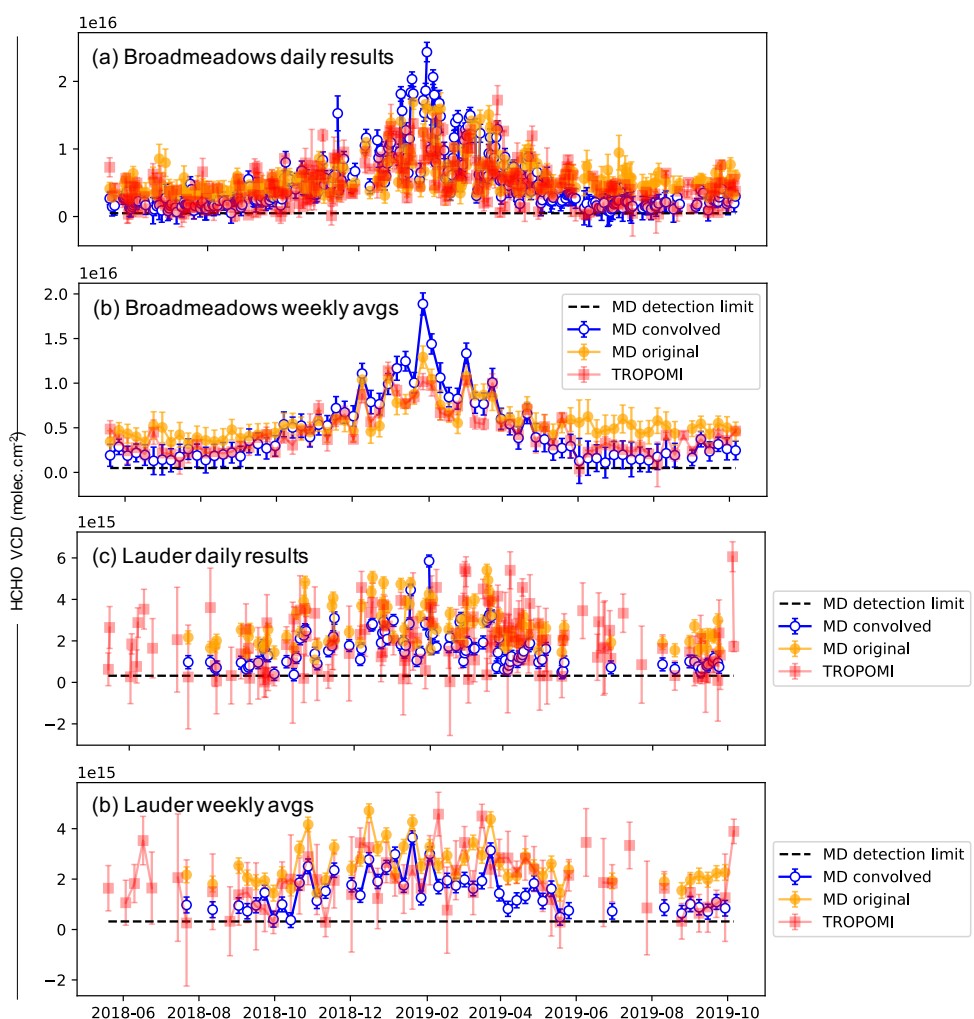

**Figure A2.** Timeseries results for the comparison of formaldehyde columns from TROPOMI and the MAX-DOAS (convolved with TROPOMI averaging kernels) between May 2018 and November 2019: (a) Broadmeadows daily measurements, (b) Broadmeadows weekly averages, (c) Lauder daily measurements and (d) Lauder weekly averages.





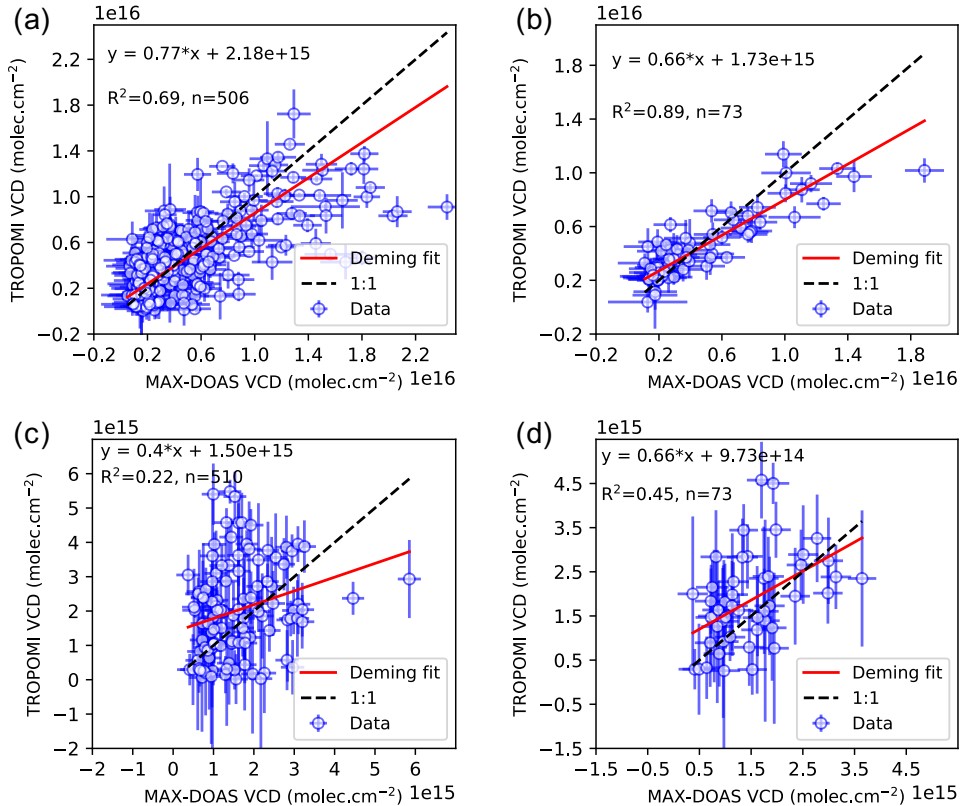

**Figure A3.** Scatter plot results for the comparison of formaldehyde columns from TROPOMI and the MAX-DOAS (convolved with TROPOMI averaging kernels) between May 2018 and November 2019: (a) Broadmeadows daily measurements, (b) Broadmeadows weekly averages, (c) Lauder daily measurements and (d) Lauder weekly averages.

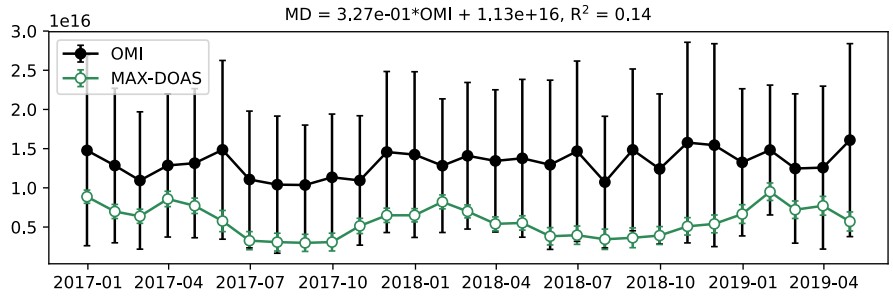

**Figure A4.** Timeseries of monthly averaged MAX-DOAS and OMI HCHO vertical columns at Broadmeadows from January 2017 to May 2019. Timeseries of monthly averaged MAX-DOAS and OMI HCHO vertical columns at Broadmeadows from January 2017 to May 2019.



**Table A1.** Results from this and previous literature studies comparing formaldehyde vertical columns from MAX-DOAS and satellite retrievals. Note that 'Diff' is for MAX-DOAS - satellite. Slope is the gradient (m) of the linear regression for Satellite = $m \times$ MAX-DOAS + C. [1] This study, [2] Tian et al. (2018), [3] Chan et al. (2019), [4] Wang et al. (2017a), [5] Hoque et al. (2018b), [6] Vigouroux et al. (2009). M = Monthly, D = Daily

| Satellite | Location | Time period | Avgs | Max. dist. (km) | Diff. $\pm$ std (%) | $R^2$ | Slope |
|---|---|---|---|---|---|---|---|
| TROPOMI[1] | Lauder, NZ | May 2018-Jan 2019 | M | 10 | -38±59 | 0.61 | 0.73 |
| TROPOMI[1] | Melbourne, Australia | May 2018-May 2019 | M | 10 | -21±34 | 0.95 | 0.61 |
| OMI[1] | Melbourne, Australia | Jan2017-May 2019 | M | 50 | -187±103 | 0.33 | 7.14 |
| OMI[2] | Yangtze R. Delta, China | Dec 2013-Dec 2016 | M | 25 | - | 0.56 | 0.84 |
| OMI[3] | Nanjing, China | Apr 2013-Apr 2017 | M | 20 | - | 0.56 | 1.01 |
| OMI[4] | Wuxi, China | May 2011-Nov 2014 | D | 50 | - | 0.17 | 2.08 |
| GOME-2[4] | Wuxi, China | May 2011-Nov 2014 | D | 50 | - | 0.18 | 1.64 |
| GOME-2[5] | Pantnagar, India | Feb-Nov 2017 | M | 50 | - | 0.50 | - |
| SCIAMACHY[6] | Reunion Island | Jul 2004-Jul 2005 | D | 500 | -11.2±30.5 | - | - |



*Author contributions.*  RGR processed the MAX-DOAS data, performed the comparison calculations, made the figures and wrote the paper. The study was designed jointly by RGR, JDS, RS, RQ and DS. JDS downloaded TROPOMI data and assisted both in comparison calculations and drafting the paper. RQ and DS were responsible for MAX-DOAS and FTIR data collection and the Lauder site and contributed to writing the manuscript. SR and MT were responsible for MAX-DOAS data collection and Broadmeadows. NJ contributed to vertical column HCHO

calculations for FTIR and MAX-DOAS results and assisted in writing the paper. RS directed the project and contributed to drafting the manuscript.

*Competing interests.*  The authors declare no competing interests exist.

*Acknowledgements.*  RS and RGR acknowledge support from the Australian Research Council's Centre of Excellence for Climate System Science (CE110001028), the Centre of Excellence for Climate Extremes (CE170100023) and Australian Research Council's Discovery

project: Tackling Atmospheric Chemistry Grand Challenges in the Southern Hemisphere (DP160101598). RR wishes to thank Prof. Peter Rayner for helpful discussions on inverse methods, Dr Udo Frieß for access to and assistance with the HEIPRO algorithm, and the Dr Albert Shimmins Memorial Fund (through the University of Melbourne) for financial support contributing to the writing of this manuscript. The authors acknowledge Thomas Danckeart, Caroline Fayt and Michel van Roozendael, authors of the QDOAS software used in DOAS analysis in this work and Blair Trewin at the Bureau of Meteorology for providing meteorological data. MAX-DOAS and FTIR measurements

at Lauder are core-funded by NIWA through New Zealand's Ministry of Business, Innovation and Employment Strategic Science Investment Fund.



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
