# Peer review of "Comparison of formaldehyde tropospheric columns in Australia and New Zealand using MAX-DOAS, FTIR and TROPOMI"

_Atmospheric Measurement Techniques, 2020_

## Referee Comment (RC1) · Anonymous Referee #1 · 30 Jul 2020

The study of Ryan et al. (2020) presents time-series of formaldehyde tropospheric columns from two ground-based MAX-DOAS stations, located in Australia and New Zealand, and their comparisons with ground-based FTIR and satellite (mainly TROPOMI) data. This paper is relevant for a publication in AMT, because the monitoring of biogenic VOCs, especially in such a region that was lacking of such measurements, is of great importance for our understanding of the complex tropospheric chemistry. The paper provides information on the observed amounts of HCHO in this region, and on the seasonal cycles, both important for modelers. The MAX-DOAS data are compared carefully with FTIR and TROPOMI data, and the paper demonstrates the improvement of the TROPOMI HCHO data compared to the OMI satellite. The paper

is also very well-structured and clear. Therefore, I recommend the publication of this paper in AMT, with a few minor comments, suggestions and questions (listed below) that should / could be addressed for some clarification.

I) Specific comments/suggestions:

- It could be nice to add a Table summarizing the main results discussed in the paper: the 2 ground-based MAX-DOAS data, with mean values, and seasonal amplitudes, and their error budget. I don't see any random and systematic uncertainty numbers given in the paper for MAX-DOAS, while I see error bars in e.g. the scatter plots (with Deming fit that is using the MAX-DOAS uncertainty). It would be clearer in a Table. Then, possibly in the same Table (or another one), the different comparisons results (slope, R2, Diff+/-Std, ErrTC,...), as in Table A1 first 3 lines: I think these lines deserve to be in the main text, not in the Appendix.

- Sect. 3.1, discussion Lauder vs Melbourne: I understand that the authors want to focus on the "background" explanation (normal conditions) of HCHO. However (suggestion), a discussion on the added possible impact of biomass burning could be interesting as well (e.g. the peak in January 2018 seen at both stations might be related to fires?). Can plumes be observed from TROPOMI this month passing over Melbourne and Lauder as well on specific days? Also, maybe a Google Earth-type map (or a vegetation map found elsewhere) could help for the discussion on the different type of vegetation (forest / agriculture,...), e.g. in replacement of Fig. 1?

- Sect.3.2, scatter plots: I do not understand the correspondence between the slope provided in the scatter plots and the red line: in Fig. A1, the slope of the red lines look larger than 1 to me, while the number given are 0.64 and .0.71. Could you explain?

- p.12, l.248: calculation of the error on the differences: in the EMAX_DOAS and EFTIR total uncertainty, do you include the smoothing error? It should be included for your comparisons of FTIR and original DOAS data (Rodgers and Connor, 2003, Eq. 13). And for comparisons of FTIR and smoothed DOAS data, then it becomes Eq.30 of

Rodgers and Connor (2003). It would be nice to have separate information on random and systematic part of the uncertainties. (of DOAS, and of the comparisons). However, the systematic part of the smoothing might be negligible in your case compared to the other sources, but it should be better to check and say it.

- p.14, l.278-298, discussion on Fig.7: may the enhanced TROPOMI HCHO columns on the east be (also) due to transport pathways from the polluted city?

- p.14, 290-292, averaging of averaging kernels: it is usually preferred to make all smoothing part before with individual data and ak, and then average the smoothed results. (cf von Clarmann, T. and Glatthor, N. , AMT, 2019). Maybe low impact in your case, did you check?

- p.16, l.318 - p.17, l.333: discussion comparisons DOAS-TROPOMI: maybe these results (slope, diff, seasonal cycle comparisons Fig.8, . . .) should be put in perspective with previous TROPOMI HCHO validation (Vigouroux et al. 2020), especially for similar stations (in HCHO amounts). Do these studies agree? Maybe the public routine validation report (not peer-reviewed) that can be found here: http://mpc-vdaf.tropomi.eu/index.php/formaldehyde can also help with MAX-DOAS / TROPOMI comparisons made at a few stations, but the sites used there are more polluted sites.

- p.17, l.340-342: the TROPOMI a priori profiles are also from chemical transport model. Did you compare the ones used for TROPOMI and for OMI retrievals? Are they so much different? A possibility to learn about the improvement made from OMI to TROPOMI would be to ask the satellite data providers (e.g. Isabelle De Smedt, BIRA-IASB).

- P.18, l. 376: "This would begin to address the current Northern-Hemispheric bias in satellite validation studies". I do not understand this sentence, sorry. Which Northern-Hemispheric bias? Which molecule? How the data in Australasia can help addressing a bias in Northern Hemisphere? Please, clarify.
II) Minor or technical comments:

- abstract, p.1, l.5 + Sect.2.3, p7, l.139 + Sect.3.1,p.9,l.193 + Sect.conclusions,p.17,l.352: the numbers given for the mean columns at Lauder are not consistent in these sections.

- abstract, p.1, l.12: I would define the partial columns.

- p.2, l.51: uses (not use)

- p.4, l.101-102: remove one of the two "however".

- p.6, legend box inserted in Fig.3(a): it is written Retrieved (blue) and Measured (Red). I guess "model" is the "Retrieved"? Maybe use the same terminology in legend of Fig.3 (and text) and the legend box in Fig.3a.

- p.6, l.129: have, not had.

- p.6, l.131: "2° the lowest possible elevation angle at Lauder": it is written 1° in the Sect. 2.1. (p.4,l.86). One should be corrected.

- p.7, l.159: The 2% random error for FTIR measurement: is it the value for the specific day 8th January 2018 or for the mean of FTIR data? It is a low value compared to the 6% given in Vigouroux et al., 2018. If this is the value for the specific day, maybe give the total column value as well or the random error in absolute values.

- p.8, l.164: "in October 2017, and is a low..." (add and ?)

- p.8, l.179: "(Gonzalez Abad et al. 2015) and are..." (add and ?)

- p.9,l.199: I would change for: "...as Melbourne is a large city, and/or to higher bio-genic VOC..." (a combination of both being likely)

- p. 12, l.245: You should specify how the percentage is calculated (mean(relative diff) or mean(absolute diff)/mean(levels)), and specify what is levels: DOAS ?FTIR? mean of both?,...

- p.17, l. 327 "The majority of. . .. line": English?

- p.18, l. 365: slope=0.61 (and not 0.81, right ?)

- p.18, l.374: "could contribute" (not could be contributed)

- p. 21, Figure A4, legend: Remove one of two repeated sentences.

---

## Referee Comment (RC2) · Anonymous Referee #2 · 4 Aug 2020

This paper presents a very interesting comparison of HCHO between two different ground-based measurement techniques: MAX-DOAS and FTIR as well as their comparison with satellite-based (TROPOMI and OMI) data. The research presented in this paper is relevant to the scientific community since in several occasions questions have been raised regarding the differences between both MAX-DOAS and FTIR measurement techniques as well as their comparison with satellite-based measurements. Moreover, this research presents novel results in the Australasian region. For these reasons I recommend the publication of this paper in AMT. A few specific and minor/technical comments are suggested below.

[Figure]

Specific comments/suggestions:

a) Page 8, line 173 and Page 9, lines 182-183: When explaining the criteria to filter satellite-based datasets it is mentioned that cloud-free data was used (cloudy scenes were avoided or excluded from the analysis), was there a threshold value used?

b) Page 21, Figure A4: When comparing ground-based measurements with OMI, which data field was used?, the "ColumnAmount" or the " ReferenceSectorCorrectedVertical-Column"?

c) Page 18, line 376: Would it be possible to please clarify the phrase " This would begin to address the current Northern-Hemispheric bias in satellite validation studies.", Is this in reference to the representativeness or the amount of monitoring stations used for satellite validations in both hemispheres?

Minor/technical comments:

Page 2, line 37: please review the phrase " Photolysis and reaction with OH and pho-tolysis"

Page 6, Figure 3a: a parenthesis is missing on the "y" axis of Figure 3a.

Page 7, line 144: a parenthesis is missing after 2018)

Page 9, line 187: cycle of formaldehyde ?

Page 10, line 2016: the forested west coast

Page 11, Figure 5: panels (b) and (c) have as insets a different panel letter (a) and (b)

Page 14, line 298: methods use different...

Page 15, line 312: well with the...

Page 17, line 327: please review the phrase "points lie are within"

Page 18, line 371: please review the phrase " will be critical tool"

---

## Referee Comment (RC3) · Anonymous Referee #3 · 5 Aug 2020

This paper presents a comparison of HCHO columns from MAX-DOAS measurements in Australia and New Zealand against ground-based FTIR and satellite measurements. The ground-based measurements consist of two novel MAX-DOAS stations located at Broadmeadows and Lauder. The MAX-DOAS measurement period consists of more than 2 years of observations, which provides information of HCHO amounts on these regions. The comparison of MAX-DOAS and FTIR at Lauder station result in a linear correlation of 0.81, while the comparison of MAX-DOAS and satellite observations at Broadmeadows and Lauder result in correlations of 0.95 and 0.61, respectively. This study provides results of HCHO at southern hemisphere where long-term observations of VOCs are missing. In addition, this study demonstrated the improvement offered by

high spatial resolution of TROPOMI measurements in comparison to previous satellite instruments.

The topic of this work fits well within the scope of AMT, the main findings are well described, the paper is well structured.

I recommend acceptance to AMT after addressing few specific comments below.

Page 4, line 93, it would be nice to have a summary table of the retrieval settings used in the HCHO retrieval

Page 4, line 100, why only these two fitting windows were selected for comparison? What is the impact of O3 absorption on the selection of HCHO fit window?

Page 5, figure 2, how is the fit error estimated?

Page 7, line 144, a parenthesis is missing after "2018)"

Page 8, line 171, what is the impact of the selection criterion o colocation distance around the station (∼10 km) in the comparison?

Page 9, line 181, why OMI use a distance of about 25 km and not 10km as TROPOMI? Would it explain the poor agreement between OMI and MAX-DOAS measurements in Figure A4, more background is captured for OMI than TROPOMI?

Page 10, line 235, although the temporal agreement is good in overall, figure 6 shown large differences from 2018-10 to 2019-01 between both dataset. Would you please discuss the possible reason of differences on the temporal variability between both datasets?

Page 13, figure 7, it would be nice to add the latitude and longitude coordinates in the figure.

Page 14, figure 8, what is the impact of HCHO amounts from fire emissions in Broadmeadows? Although, the station is located on the city and HCHO, however in the

last years many fires have been observed in southeast of Australia, which could emit and transport HCHO from far location similar to observed in recent study for Canada wildfires.

Page 14, line 298, please replace "methods usec" by "methods use"

Page 21, figure A4, the low spatial resolution of OMI in comparison to TROPOMI could be a reason for the not variability of HCHO. Would you expect a similar result for instruments with morning overpass like GOME2 and lower spatial resolution than OMI?

---

## Author Comment (AC1) · 27 Aug 2020

Reviewer comments in black text
*Author responses in orange italic text*
*Additions to the manuscript in blue italic text*

*Authors' note:*
*An error was found in the time-matching of FTIR and MAX-DOAS columns, so the comparison plots (figures 6 and A1) have been updated accordingly.*

**Anonymous Referee #1**

The study of Ryan et al. (2020) presents time-series of formaldehyde tropospheric columns from two ground-based MAX-DOAS stations, located in Australia and New Zealand, and their comparisons with ground-based FTIR and satellite (mainly TROPOMI) data. This paper is relevant for a publication in AMT, because the monitoring of biogenic VOCs, especially in such a region that was lacking of such measurements, is of great importance for our understanding of the complex tropospheric chemistry. The paper provides information on the observed amounts of HCHO in this region, and on the seasonal cycles, both important for modelers. The MAX-DOAS data are compared carefully with FTIR and TROPOMI data, and the paper demonstrates the improvement of the TROPOMI HCHO data compared to the OMI satellite. The paper is also very well-structured and clear. Therefore, I recommend the publication of this paper in AMT, with a few minor comments, suggestions and questions (listed below) that should / could be addressed for some clarification.

I) Specific comments/suggestions:

- It could be nice to add a Table summarizing the main results discussed in the paper: the 2 ground-based MAX-DOAS data, with mean values, and seasonal amplitudes, and their error budget. I don't see any random and systematic uncertainty numbers given in the paper for MAX-DOAS, while I see error bars in e.g. the scatter plots (with Deming fit that is using the MAX-DOAS uncertainty). It would be clearer in a Table. Then, possibly in the same Table (or another one), the different comparisons results (slope, R2, Diff+/-Std, ErrTC,. . .), as in Table A1 first 3 lines: I think these lines deserve to be in the main text, not in the Appendix.
*– The following text has been added in Section 2.3 by way of overview of random and systematic errors in the MAX-DOAS process:*

*"The errors associated with the MAX-DOAS retrieval include systematic errors, which derive primarily from HCHO cross section uncertainty which is around 9% (Vigouroux et al., 2009). Random errors include model parameter uncertainty (such as uncertainty in a priori parameters), estimated to be 10% following the methodology outlined in Ryan et al. (2018), along with retrieval noise and smoothing errors which were calculated in HEIPRO."*

*We have added the first table requested in Section 3.1, summarising the random, systematic and overall errors along with the mean HCHO column and seasonal amplitude at each MAX-DOAS measurement site. We have also added the requested comparison summary table incorporating all relevant statistics and error breakdown.*

- Sect. 3.1, discussion Lauder vs Melbourne: I understand that the authors want to focus on the "background" explanation (normal conditions) of HCHO. However (suggestion), a discussion on the added possible impact of biomass burning could be interesting as well (e.g. the peak in January 2018 seen at both stations might be related to fires?). Can plumes be observed from TROPOMI this month passing over Melbourne and Lauder as well on specific days? Also, maybe a Google Earth-type map (or a vegetation map found elsewhere) could help for the discussion on the different type of vegetation (forest / agriculture,. . .), e.g. in replacement of Fig. 1*? – This is a good point; biomass burning is also likely to be a large source of oxidised VOCs especially in south-eastern Australia. We have included at line 215 the following (following from noting that the source directions of HCHO in Melbourne were from the north and east, the directions of the the most proximate forests): "The dominant source directions from forested and rural regions, along with summer time peak, are also consistent with biomass burning being a source of formaldehyde in Melbourne."*

*We have updated Figure 1 to Google Earth-derived maps which allow vegetation to be seen, and aids in the interpretation of source directions.*

*The idea of using TROPOMI to study smoke plumes extending from south-eastern Australia, including over New Zealand, is a good one especially given the recent (2019-20) large fire events. We believe this work could comprise a paper all of its own and is outside the scope of this comparison study.*

- Sect.3.2, scatter plots: I do not understand the correspondence between the slope provided in the scatter plots and the red line: in Fig. A1, the slope of the red lines look larger than 1 to me, while the number given are 0.64 and .0.71. Could you explain? *– thanks for alerting me to this error. The linear least squares regression equations have been printed by mistake, rather than the Deming regression equation (which is plotted). They have been updated now.*

- p.12, l.248: calculation of the error on the differences: in the EMAX_DOAS and EFTIR total uncertainty, do you include the smoothing error? It should be included for your comparisons of FTIR and original DOAS data (Rodgers and Connor, 2003, Eq. 13). And for comparisons of FTIR and smoothed DOAS data, then it becomes Eq.30 of C2 Rodgers and Connor (2003). It would be nice to have separate information on random and systematic part of the uncertainties. (of DOAS, and of the comparisons). However, the systematic part of the smoothing might be negligible in your case compared to the other sources, but it should be better to check and say it. *– yes, the smoothing error is incorporated in the errors used to calculate the error on the differences. Following from this reviewer comment, the calculation of the differences of the raw and smoothed columns, along with breakdown of random and systematic errors on the differences, have been carried out using the methodology in Vigouroux 2009 and inserted into the tables requested above.*

- p.14, l.278-298, discussion on Fig.7: may the enhanced TROPOMI HCHO columns on the east be (also) due to transport pathways from the polluted city?
*The largest cities in the South Island lie to the south east and north east of Lauder, rather than directly to the east. Nevertheless, the point is in general terms a good one because the population density is highest along the east coast of the South Island. This has now been noted in the discussion of Fig. 5 on page 10:*

*"The highest population density in the South Island, including the cities of Dunedin and Christchurch, lies along the east coast. Given that the lifetime of formaldehyde is on the order of hours, transport on the order of a hundred kilometres is possible meaning the different source directions can reasonably be compared. On the available evidence it could be hypothesised that the agricultural and more densely populated eastern sector is a stronger source of formaldehyde to Lauder than the forested the west coast".*

*Also noted on page 10:*

*"...figure 7(b) supports the conclusion (from the MAX-DOAS measurements) that the highest formaldehyde amounts are in the agricultural and more densely populated eastern parts of the island."*

- p.14, 290-292, averaging of averaging kernels: it is usually preferred to make all smoothing part before with individual data and ak, and then average the smoothed results. (cf von Clarmann, T. and Glatthor, N. , AMT, 2019). Maybe low impact in your case, did you check? *– thank you for alerting me to this interesting point. I had not checked initially, but now have, and find that the correct smoothing approach does not change the regression or bias results of the comparison.*

- p.16, l.318 - p.17, l.333: discussion comparisons DOAS-TROPOMI: maybe these results (slope, diff, seasonal cycle comparisons Fig.8, . . .) should be put in perspective with previous TROPOMI HCHO validation (Vigouroux et al. 2020), especially for similar stations (in HCHO amounts). Do these studies agree? Maybe the public routine validation report (not peer-reviewed) that can be found here: http://mpcvdaf.tropomi.eu/index.php/formaldehyde can also help with MAX-DOAS / TROPOMI comparisons made at a few stations, but the sites used there are more polluted sites. *– yes, good point, following from the revised calculation of column differences, comparisons have been made with Vigouroux 2020 (which is also the most pertinent reference for validation in the TROPOMI public routine validation report!).*

- p.17, l.340-342: the TROPOMI a priori profiles are also from chemical transport model. Did you compare the ones used for TROPOMI and for OMI retrievals? Are they so much different? A possibility to learn about the improvement made from OMI to TROPOMI would be to ask the satellite data providers (e.g. Isabelle De Smedt, BIRA-IASB). *– Good point, in fact according to De Smedt et al., 2018, the HCHO a priori profiles are calculated using the same chemical transport model in both OMI and TROPOMI retrievals. The key difference between the instruments lies in the resolution improvement. The text has been revised as follows:*

*Previous text: "Another possibility (for discrepancy between OMI and TROPOMI comparisons) could be inappropriate a priori profiles, which are calculated in OMI using chemical transport models. Knowing that formaldehyde production from isoprene and monoterpene emissions is poorly simulated in south-eastern Australia using standard VOC mechanisms (Emmerson et al., 2016, 2018), relying on these mechanisms to produce useful formaldehyde a priori profiles in this area may be problematic. Previous studies (e.g. De Smedt et al., 2015; Wang et al.,2017a) found that agreement between OMI and MAX-DOAS measurements was found when using the MAX-DOAS a priori profiles to retrieve satellite columns; it would be interesting in future work to do the same for HCHO satellite-based retrievals over Australasia."*

*Revised text: "Given that both OMI and TROPOMI retrievals rely on a priori formaldehyde profiles calculated using the same chemical transport model (TM5, De Smedt 2018), a priori differences cannot explain the difference in the comparison. However, revious studies (e.g. De Smedt et al., 2015; Wang et al.,2017a) found that agreement between OMI and MAX-DOAS measurements improved when using the MAX-DOAS a priori profiles to retrieve satellite columns; it would be interesting in future work to do the same for HCHO satellite-based retrievals over Australasia. Examining the influence of a priori profiles calculated by chemical transport models on formaldehyde retrievals is also of particular interest in south-eastern Australia given that biogenic VOC emissions have been shown to be poorly simulated in this region (Emmerson et al., 2016, 2018)."*

- P.18, l. 376: "This would begin to address the current Northern-Hemispheric bias in satellite validation studies". I do not understand this sentence, sorry. Which Northern Hemispheric bias? Which molecule? How the data in Australasia can help addressing a bias in Northern Hemisphere? Please, clarify. *– We refer here to a sampling bias rather than a numerical bias, that is, the lack of satellite validation results in the Southern Hemisphere. The sentence in question has been revised to "This would continue to address the lack of Southern-Hemispheric satellite validation studies using ground-based remote sensing."*

II) Minor or technical comments:

- abstract, p.1, l.5 + Sect.2.3, p7, l.139 + Sect.3.1,p.9,l.193 + Sect.conclusions,p.17,l.352: the numbers given for the mean columns at Lauder are not consistent in these sections. *– The correct value of   molecules cm$^{-2}$ has been updated.*

- abstract, p.1, l.12: I would define the partial columns. *– fixed, now reads "... partial columns (0-4 km)."*

- p.2, l.51: uses (not use) *– fixed*

- p.4, l.101-102: remove one of the two "however". *– fixed*

- p.6, legend box inserted in Fig.3(a): it is written Retrieved (blue) and Measured (Red). I guess "model" is the "Retrieved"? Maybe use the same terminology in legend of Fig.3 (and text) and the legend box in Fig.3a. *– fixed, Fig. 3(a) caption now reads "shows retrieved-measured dSCD comparison"*

- p.6, l.129: have, not had. *– fixed*

- p.6, l.131: "2∘ the lowest possible elevation angle at Lauder": it is written 1∘ in the Sect. 2.1. (p.4,l.86). One should be corrected. *– lowest is 2º, p.4 l.86 has been corrected*

- p.7, l.159: The 2% random error for FTIR measurement: is it the value for the specific day 8th January 2018 or for the mean of FTIR data? It is a low value compared to the 6% given in Vigouroux et al., 2018. If this is the value for the specific day, maybe give the total column value as well or the random error in absolute values. *– This value is the mean for the FTIR data.*

- p.8, l.164: "in October 2017, and is a low. . ." (add and ?)  *- fixed*

- p.8, l.179: "(Gonzalez Abad et al. 2015) and are. . ." (add and ?) *– in fact I have removed the "are" from earlier in the sentence to make it read more clearly.*

- p.9,l.199: I would change for: ". . .as Melbourne is a large city, and/or to higher biogenic VOC. . ." (a combination of both being likely)  *- fixed*

- p. 12, l.245: You should specify how the percentage is calculated (mean(relative diff) or mean(absolute diff)/mean(levels)), and specify what is levels: DOAS ?FTIR? mean of both

- p.17, l. 327 "The majority of. . .. line": English?  *- fixed ("are" removed).*

- p.18, l. 365: slope=0.61 (and not 0.81, right ?) *– yes, typo corrected.*

- p.18, l.374: "could contribute" (not could be contributed) *– fixed*

- p. 21, Figure A4, legend: Remove one of two repeated sentences *– fixed*

---

## Author Comment (AC3) · 28 Aug 2020

amt-2020-232 – Interactive discussion Reviewer comments in black text Author responses in orange italic text Additions to the manuscript in blue italic text

**Anonymous Referee #3 Received and published: 5 August 2020**

This paper presents a comparison of HCHO columns from MAX-DOAS measurements in Australia and New Zealand against ground-based FTIR and satellite measurements. The ground-based measurements consist of two novel MAX-DOAS stations located at Broadmeadows and Lauder. The MAX-DOAS measurement period consists of more than 2 years of observations, which provides information of HCHO amounts on these regions. The comparison of MAX-DOAS and FTIR at Lauder station result in a linear correlation of 0.81, while the comparison of MAX-DOAS and satellite observations at Broadmeadows and Lauder result in correlations of 0.95 and 0.61, respectively. This study provides results of HCHO at southern hemisphere where long-term observations of VOCs are missing. In addition, this study demonstrated the improvement offered by high spatial resolution of TROPOMI measurements in comparison to previous satellite instruments. The topic of this work fits well within the scope of AMT, the main findings are well described, the paper is well structured. I recommend acceptance to AMT after addressing few specific comments below.

Page 4, line 93, it would be nice to have a summary table of the retrieval settings used in the HCHO retrieval - We have decided to list retrieval settings for all instruments and processes (MAX-DOAS spectral settings, MAX-DOAS profile retrieval, FTIR and TROPOMI retrieval) rather than employ tables. With respect, we feel that the DOAS spectral settings are summarised clearly and succinctly enough in the text as is.

Page 4, line 100, why only these two fitting windows were selected for comparison? What is the impact of O3 absorption on the selection of HCHO fit window? – *As noted in lines 98-99 and in more detail in lines 102-105, the two bands were chosen because they are the two most commonly used in the literature. This is because of a trade-off between incorporating five HCHO absorption bands in the fit (324.5-359 nm) with more chance of spectral interference from ozone, or only three HCHO bands in the fit (336-359 nm). The aim of the sensitivity study was to determine whether there was any improvement in the fit possible from including more spectral information.*

While  $O_3$  absorption is undoubtedly more significant going towards the UV end of the wavelength ranges, any concern that extra  $O_3$  absorption is problematic in the 324.5-359 range is alleviated by the fact that neither the residual RMS nor HCHO dSCD (figure 2)) are impacted in this sensitivity test.

Page 5, figure 2, how is the fit error estimated? – Thank you for this question because it allows a more insightful analysis of why the 324.5-359 nm fitting range gave a lower fit error. The fit error quoted in Fig. 2 is the fit error calculated in QDOAS, resulting from the covariance matrix  $E_x$  of the optical density fit result. This is estimated by:

 $E_{\chi} = \chi^{2} (A^{T} A)^{-1}$ where A contains the linear components of the fit and  $\chi^{2} = \frac{\sum_{i=1}^{M} (f^{i})^{2}}{M - N}$

with  $(f^i)^2$  being the sum of squares of the residuals, M being the number of wavelengths included in the fit and N being the number of fitted parameters. Given that we know neither the result of the fit (Fig. 2(a)) nor residuals (Fig. 2(c)) were substantially impacted by the choice of wavelength range we can conclude that the improvement in fit error comes from the M-N term, i.e. the information content of the retrieval. In other words, more information is supplied to the retrieval using the longer wavelength range with no accompanying deterioration in the residual RMS. The text has been modified to make this point.

Original text: "As shown in figure 2, the change of wavelength range had minimal impact on the residual RMS throughout the day, and minimal impact on the magnitude of the HCHO dSCD. However, despite minimal change in dSCD, the fit error was greatly reduced by using the longer wavelength range, therefore the 324.5-359 nm range was adopted as the standard employed for fitting formaldehyde"

*Modified text:* "The calculation of fit error in QDOAS depends on the linear fit parameters, the residuals and the information content of the retrieval, which depends in turn on the number of wavelengths in the fit. Neither the residual RMS (Fig. 2(c)) nor magnitude of the dSCD (Fig. 2(a)) were substantially impacted by the choice of wavelength range, suggesting that the improvement in fit error for the 324.5-359 nm range (Fig. 2(b)) results from increasing the information content of the retrieval. As a result of the increased information content and resulting lower fit errors, the 324.5-359 nm range was adopted in this paper for formaldehyde."

Page 7, line 144, a parenthesis is missing after "2018)" - fixed

Page 8, line 171, what is the impact of the selection criterion o colocation distance around the station (~10 km) in the comparison? – The comparison statistics do change for different TROPOMI resolutions sampled around the measurement location. We have re-run the comparisons at different sampling resolutions and added the following sentences in Section 3.3:

"The resolution selection criterion did not have a large effect on the comparison, with a regression slope of 0.68 (monthly averages) for averaging TROPOMI 50 km either side of Lauder as opposed to 20 km."

"Considering TROPOMI sampled 10km and 50 km either side of Broadmeadows, regression slopes were 0.56 and 0.65 respectively, with the low bias of TROPOMI compared to MAX-DOAS at high HCHO consistent across sampling resolution"

Page 9, line 181, why OMI use a distance of about 25 km and not 10km as TROPOMI? Would it explain the poor agreement between OMI and MAX-DOAS measurements in Figure A4, more background is captured for OMI than TROPOMI? – As noted in section 2.5, the quoted spatial resolution of OMI is 13x24 km, compared to TROPOMI  $3.6 \times 7.2$  km (before 6 August 2019) and  $3.6 \times 5.6$  km (after 6 August 2019). This is the reason why the OMI sampling for analysis was 25 km. However, an error was made in the manuscript with TROPOMI columns actually compared at 0.2 degree resolution rather than 0.1 degree owing to the small number of pixels per cell at Lauder. This has been amended in the manuscript. For the OMI comparison, OMI will sample a little more of the background but we do not consider that this effect should remove the ability to see the HCHO seasonality – it doesn't for TROPOMI at 25 km either side of the measurement location. We have added the following in the discussion of Fig. A4 at the end of Section 3.3: "...the fact that OMI is sampled 25 km either side of the measurement location compared to approximately 20 km for MAX-DOAS, thereby taking in more of the background. However, this could not explain why no seasonality is evident in the OMI results."

Page 10, line 235, although the temporal agreement is good in overall, figure 6 shown large differences from 2018-10 to 2019-01 between both dataset. Would you please discuss the possible reason of differences on the temporal variability between both datasets? – *This is a good question. We cannot provide a definitive answer for the difference in terms of HCHO sources because there is limited information on formaldehyde in NZ. However, given the sensitivities of the instruments, we suspect that the difference between the agreement in summer between the instruments lies in lower HCHO plumes in summers 2016-17 and 2018-19 which can be better detected by the MAX-DOAS than FTIR. We have updated the text describing this in Section 3.2:*

*Previous: "The month-to-month variation in formaldehyde is in especially good temporal agreement for summer (DJF) 2017-18, while the summer increase is less clearly captured by the FTIR in summer 2016-17 or 2018-19."*

*Revised:* "The month-to-month variation in formaldehyde is in especially good temporal agreement for summer (DJF) 2017-18, while both the timing and magnitude of HCHO in summer 2016-17 and 2018-19 are poorly replicated by the FTIR. Due to the higher sensitivity of the MAX-DOAS to the lower troposphere this suggests HCHO plumes were lower in 2016-17 and 2018-19, and therefore not captured as well by the FTIR, than summer 2017-18."

Page 13, figure 7, it would be nice to add the latitude and longitude coordinates in the figure. – *this has been done*.

Page 14, figure 8, what is the impact of HCHO amounts from fire emissions in Broadmeadows? Although, the station is located on the city and HCHO, however in last years many fires have been observed in southeast of Australia, which could emit and transport HCHO from far location similar to observed in recent study for Canada wildfires. – good point, indeed we expect biomass burning should be a source of HCHO in Melbourne. We have discussed biomass burning above in the responses to Reviewer 1.

Page 14, line 298, please replace "methods usec" by "methods use" - *fixed*

Page 21, figure A4, the low spatial resolution of OMI in comparison to TROPOMI could be a reason for the not variability of HCHO. Would you expect a similar result for instruments with morning overpass like GOME2 and lower spatial resolution than OMI? – this follows on from the previous point about Fig. A4 – yes, the lower spatial resolution of OMI may help blur the seasonal variation somewhat. However, TROPOMI and the MAX-DOAS show a strong seasonality at this location and also a strong dependence of HCHO on wind direction (indicating there are clearly strong HCHO sources in the region). Therefore I am surprised at the lack of seasonality in OMI results. Regarding GOME2, the diurnal variation of HCHO at this location peaks in the middle of the day and afternoon, so it would be interesting to see whether the morning overpass would also make it difficult to observe strong seasonality. This

is outside the scope of the present study however, which is primarily to compare MAX-DOAS with FTIR and TROPOMI.

---

## Author Comment (AC2)

Reviewer comments in black text
*Author responses in orange italic text*
*Additions to the manuscript in blue italic text*

**Anonymous Referee #2**

This paper presents a very interesting comparison of HCHO between two different ground-based measurement techniques: MAX-DOAS and FTIR as well as their comparison with satellite-based (TROPOMI and OMI) data. The research presented in this paper is relevant to the scientific community since in several occasions questions have been raised regarding the differences between both MAX-DOAS and FTIR measurement techniques as well as their comparison with satellite-based measurements. Moreover, this research presents novel results in the Australasian region. For these reasons I recommend the publication of this paper in AMT. A few specific and minor/technical comments are suggested below.

Specific comments/suggestions:

a) Page 8, line 173 and Page 9, lines 182-183: When explaining the criteria to filter satellite-based datasets it is mentioned that cloud-free data was used (cloudy scenes were avoided or excluded from the analysis), was there a threshold value used? *– Cloudy scenes were determined by filtering for the recommended quality control parameter < 0.5, which incorporates cloud radiance fraction (at 340 nm) < 0.5. The relevant text on page 8 has been updated:*

*"The recommended quality control (QC) filtering was applied, excluding retrieved values where the QC flag was less than 0.5 (on a scale of 0-1), which ensures scenes with cloud radiance fraction (at 340 nm) < 0.5 are excluded from the comparisons."*

b) Page 21, Figure A4: When comparing ground-based measurements with OMI, which data field was used?, the "ColumnAmount" or the " ReferenceSectorCorrectedVerticalColumn"? *– The "ColumnAmount" field was used. This has now been noted in the figure caption of A4.*

c) Page 18, line 376: Would it be possible to please clarify the phrase " This would begin to address the current Northern-Hemispheric bias in satellite validation studies.", Is this in reference to the representativeness or the amount of monitoring stations used for satellite validations in both hemispheres? *– Yes, a sampling bias of SH vs NH validations studies – see response to the same question from Reviewer 1.*

Minor/technical comments:

Page 2, line 37: please review the phrase "Photolysis and reaction with OH and photolysis" *– amended to "Photolysis and reaction with OH limit…"*

Page 6, Figure 3a: a parenthesis is missing on the "y" axis of Figure 3a. *– fixed*

Page 7, line 144: a parenthesis is missing after 2018) *– fixed*

Page 9, line 187: cycle of formaldehyde? *– fixed*

Page 10, line 216: the forested west coast *– fixed*

Page 11, Figure 5: panels (b) and (c) have as insets a different panel letter (a) and (b) *– fixed*

Page 14, line 298: methods use different... *– fixed*

Page 15, line 312: well with the... *– fixed*

Page 17, line 327: please review the phrase "points lie are within" *– fixed, "points lie within"*

Page 18, line 371: please review the phrase "will be critical tool" *– fixed, "will be a critical tool"*

---

## Author Response (AR2)

**Author's response to request for minor revisions**

The Associate Editor raised the following points:

**Associate Editor Decision: Publish subject to minor revisions (review by editor)** (20 Oct 2020) by Michel Van Roozendael
Comments to the Author:
The comments raised by reviewers, are very well covered in your detailed answers, however I have one more comment and a question:
(1) Fig.3: the usual convention is to refer to 'measured' for the observed values (dSCDs derived from the DOAS fit) and 'calculated' for the simulated values (obtained using Heipro).
(2) Fig.4: the local maximum at 2 km in the FTIR averaging kernel is quite surprising (at least to me). I was expecting to see a monotonic increase in sensitivity up to approx. 10km, as e.g. reported in Vigouroux et al. (2009). Please could you comment on the origin of this local maximum?

We thank the Associate Editor for his decision and questions.

(1) We have updated the label of Fig. 3(a) to reflect the convention
(2) The averaging kernels in Fig.4 looked odd because an error was made in plotting (columns and rows mixed up). We sincerely apologise for this error which we should have picked up on earlier. The correct averaging kernels have now been plotted in Fig. 4 and much more closely resemble the results in Vigouroux et al. (2009).